# Modeling spatial contrast sensitivity in responses of primate retinal ganglion cells to natural movies

Shashwat Sridhar[1,2*], Michaela Vystrčilová[3], Mohammad H. Khani[1,2¤a], Dimokratis Karamanlis[1,2¤b], Helene M. Schreyer[1,2¤a], Varsha Ramakrishna[1,2,4], Steffen Krüppel[1,2], Sören J. Zapp[1,2], Matthias Mietsch[5,6], Alexander S. Ecker[2,3,7,8], Tim Gollisch[1,2,8,9*]

**1** Department of Ophthalmology, University Medical Center Göttingen, Göttingen, Germany, **2** Bernstein Center for Computational Neuroscience Göttingen, Göttingen, Germany, **3** University of Göttingen, Institute of Computer Science and Campus Institute Data Science, Göttingen, Germany, **4** International Max Planck Research School for Neurosciences, Göttingen, Germany, **5** German Primate Center, Laboratory Animal Science Unit, Göttingen, Germany, **6** German Center for Cardiovascular Research, Partner Site Göttingen, Göttingen, Germany, **7** Max Planck Institute for Dynamics and Self-Organization, Göttingen, Germany, **8** Cluster of Excellence "Multiscale Bioimaging: from Molecular Machines to Networks of Excitable Cells" (MBExC), University of Göttingen, Göttingen, Germany, **9** Else Kröner Fresenius Center for Optogenetic Therapies, University Medical Center Göttingen, Göttingen, Germany

¤a Current Address: Institute of Molecular and Clinical Ophthalmology Basel, Basel, Switzerland
¤b Current Address: University of Geneva, Department of Basic Neurosciences, Geneva, Switzerland
* shashwat.sridhar@med.uni-goettingen.de (SS); tim.gollisch@med.uni-goettingen.de (TG)

## Abstract

Retinal ganglion cells, the output neurons of the vertebrate retina, often display non-linear summation of visual signals over their receptive fields. This creates sensitivity to spatial contrast, letting the cells respond to spatially structured visual stimuli even when no net change in overall illumination of the receptive field occurs. Yet, computational models of ganglion cell responses are often based on linear receptive fields, and typical nonlinear extensions, which separate receptive fields into nonlinearly combined subunits, are often cumbersome to fit to experimental data. Previous work has suggested to model spatial-contrast sensitivity in responses to flashed images by combining signals from the mean and variance of light intensity inside the receptive field. Here, we extend and adjust this spatial contrast model for application to spatiotemporal stimulation and explore its performance on spiking responses that we recorded from ganglion cells of marmosets under artificial and naturalistic movies. We show how the model can be fitted to experimental data and that it outperforms common models with linear spatial integration to different degrees for different types of ganglion cells. Finally, we use the model framework to infer the cells' spatial scale of nonlinear spatial integration. Our work shows that the spatial contrast model can capture aspects of nonlinear spatial integration in the primate retina with only few free parameters. The model can be used to assess the cells' functional properties under natural stimulation and provides a simple-to-obtain benchmark for comparison with more detailed nonlinear encoding models.

which permits unrestricted use, distribution, and reproduction in any medium, provided the original author and source are credited.

**Data availability statement:** The data of this study have been made publicly available on G-Node (https://doi.org/10.12751/g-node.t43ph1) together with information for reconstructing the visual stimuli. The code for fitting and evaluating the models has been made available on GitHub (https://github.com/gollischlab/SpatiotemporalSCModel).

**Funding:** This work was supported by the Deutsche Forschungsgemeinschaft (DFG, German Research Foundation) – project numbers 432680300 (SFB 1456, project B05) to TG and ASE, 528760423 (SFB 1690/1, project B05) to TG, and 515774656 to TG – and by the European Research Council (ERC) under the European Union's Horizon 2020 research and innovation programme (grant agreement number 101041669 to ASE). The funders had no role in study design, data collection and analysis, decision to publish, or preparation of the manuscript.

**Competing interests:** The authors have declared that no competing interests exist.

## Author summary

Our visual experience depends on the retina's remarkable ability to detect light patterns and contrast in the world around us. Retinal ganglion cells, the output neurons of the retina, modulate their activity based on signals within small, specific regions of the visual scene, called their receptive fields. Many of these cells not only encode overall brightness, summed linearly across the receptive field, but are also sensitive to local spatial contrast, that is, variations in brightness within the receptive field. Computational models that account for this nonlinear spatial integration exist, but require large amounts of data and are challenging to fit. We therefore developed the spatial contrast model, which takes a simple measure of light-intensity variations as an input, and tested it on measured responses of primate retinal ganglion cells to both artificial and naturalistic movies. The model substantially outperformed standard models with linear receptive fields, despite having only one additional tunable parameter. Furthermore, we used the model to investigate the spatial scale at which the cells integrate spatial contrast and found striking consistency across cell types. The spatial contrast model thus offers a practical tool for capturing retinal stimulus encoding and a simple-to-obtain benchmark for modeling nonlinear spatial integration.

## Introduction

The ganglion cells of the vertebrate retina encode visual stimuli that occur inside their receptive fields. When they are activated by an appropriate stimulus, the cells report this by generating spikes for transmission along their axons, the fibers of the optic nerve. The most basic principle of this encoding is that ON ganglion cells are activated by an increase of light intensity in their receptive field centers, OFF ganglion cells by a decrease of light intensity, and ON-OFF cells by light-intensity changes of either sign [1]. In computational models of retinal ganglion cell activity, this is often captured by treating the receptive field as a (linear) filter of the incoming light-intensity pattern. Applying this filter amounts to computing a weighted spatial summation of the change in illumination, with weights given by the profile of the receptive field over space. The obtained filter activation can then be transformed into an actual response, such as a firing rate, in a way that takes the preferred contrast polarity of the ganglion cell — ON, OFF, or ON-OFF — into account. In its most basic form, this procedure corresponds to the commonly used linear–nonlinear (LN) model [2], where the receptive-field filtering represents the model's linear stage and the subsequent transformation into the cell's response the model's nonlinear stage.

Yet, despite the success of the LN model in describing retinal ganglion cell responses, as well as of its many variants that keep the linear receptive field for spatial signal integration, it has long been known that many ganglion cells display nonlinear receptive fields [3]. Typically, this is demonstrated by responses to stimulation with contrast-reversing spatial gratings, showing that gratings can evoke sizable

responses even when the contrast reversals do not activate the linear receptive field because light-intensity increments and decrements symmetrically balance each other. The nonlinearities of the receptive field then lead to a characteristic frequency doubling in the response, as contrast reversals in both directions can activate the cell about equally strongly.

The receptive-field nonlinearities are, for some ganglion-cell types, considered a crucial component to be included in computational models for accurately capturing responses to natural stimuli [4–9]. A typical approach is to separate the receptive field into multiple smaller spatial filters, so-called subunits, and to then nonlinearly combine the resulting filter signals [10]. Several methods for inferring subunits from experimental data have been suggested (e.g., [11–16]), which, in addition to serving as the basis for modeling, may also provide insights into retinal circuitry, as the subunits are thought to correspond to the receptive fields of presynaptic bipolar cells [17–19]. Yet, subunit inference remains a formidable challenge, requiring long recordings and computationally demanding analyses. Moreover, it is not clear how much the actual precise subunit locations matter for response predictions [20].

As a simpler alternative to subunit models, one may therefore consider capturing nonlinear spatial integration by including information about higher-order statistics of light intensities inside the receptive field. Indeed, prior work on modeling responses from salamander and mouse retinal ganglion cells to flashed natural images has shown that a substantial improvement over the LN model can be obtained by simply providing a measure of spatial contrast inside the receptive field as an additional input channel [21]. Here, we extend and modify this spatial contrast model to also include temporal integration, thus allowing application to spatiotemporal stimulation. Moreover, we apply the model to data recorded from marmoset retinal ganglion cells and demonstrate that the concept of including spatial contrast information is useful also for modeling the primate retina. In response to artificial and natural movies, we find that the spatial contrast model can outperform the LN model in a cell-type-dependent and stimulus-dependent manner and that it is not surpassed in performance by a standard subunit model, despite the latter's detail in modeling spatial nonlinearities. In addition, we use the model to probe the ganglion cells' scale of spatial contrast integration, which separates linear integration of small spatial features from nonlinear integration and sensitivity to spatial structure larger than this scale. By introducing a smoothing step in the spatial contrast model after temporal filtering, we find that, across cell types, this spatial scale falls within a narrow range, which corresponds to known sizes of bipolar cell dendritic trees in the marmoset.

## Results

### Spatial-contrast-driven nonlinear integration in retinal ganglion cells

To investigate how spatial contrast inside the receptive fields of primate retinal ganglion cells affects stimulus encoding under natural movies, we recorded the spiking activity from marmoset ganglion cells in response to various stimuli with microelectrode arrays. To be able to assess the stimulus inside a cell's receptive field, we used reverse-correlation analysis under stimulation with spatiotemporal white noise to determine, for each recorded cell, its spatial receptive field and temporal filter.

In order to study cell-type-dependent differences, we assigned the cells to clusters based on their receptive-field sizes, their temporal filters, and the autocorrelograms of their spike trains (see *Methods*). We focused on four classes of cells: OFF midget cells, OFF parasol cells, ON parasol cells, and Large OFF cells (Fig 1A). OFF midget, OFF parasol, and ON parasol cells were obtained as well-separated clusters, each with consistent autocorrelograms and temporal filters and with receptive fields of similar sizes, which approximately tiled the retinal surface (despite gaps in the mosaics from cells which were not recorded). Parasol cells were distinguished from midget cells by their larger receptive fields and faster temporal filters (Fig 1B), as expected [11,22]. The Large OFF cells in our analysis were a collection of cells from several clusters with somewhat more heterogeneous properties and likely belonged to several cell types, which were nonetheless pooled together here, as the boundaries between the contributing clusters were less clear than for midget and parasol cells.

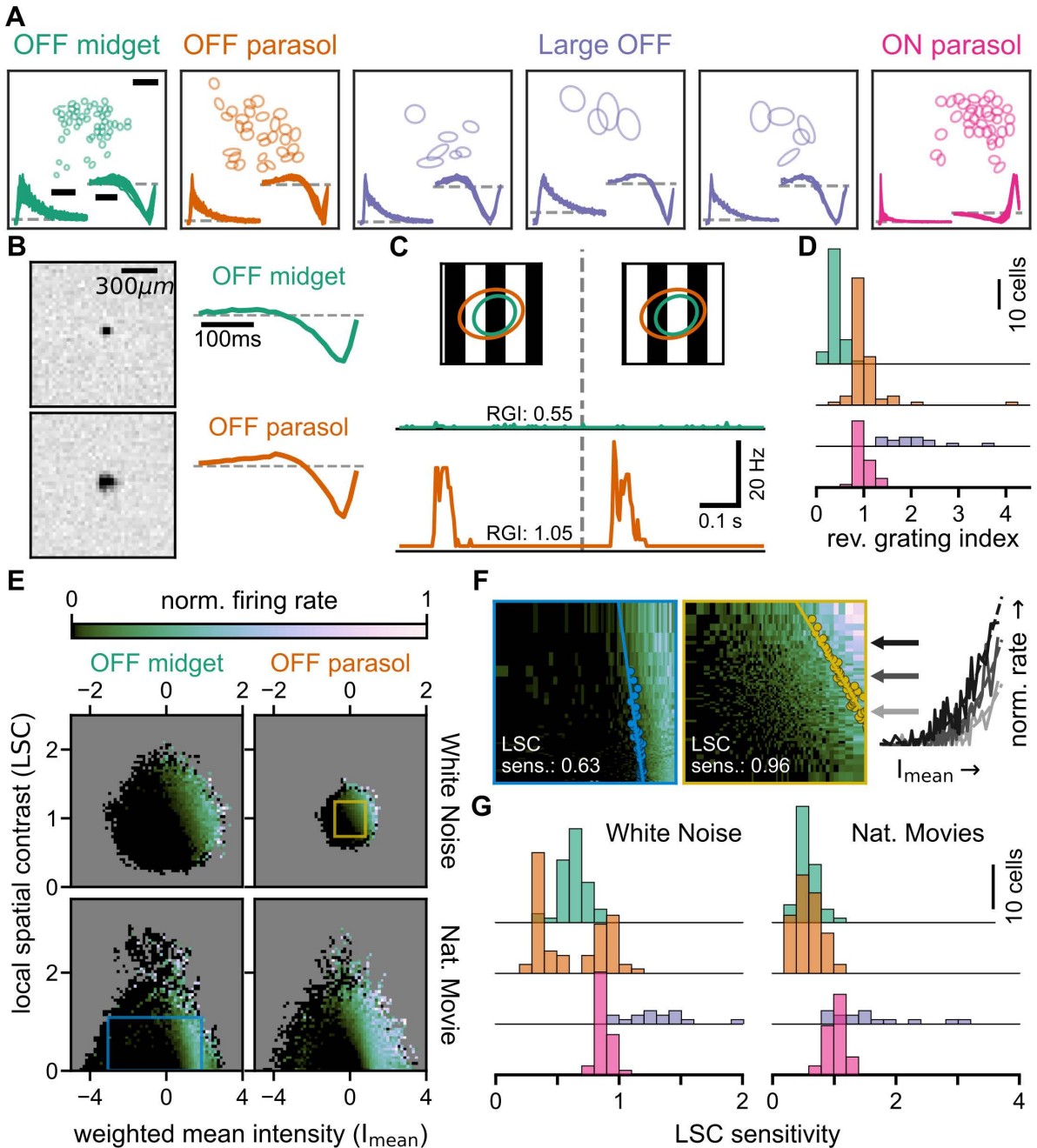

**Fig 1. Spatial receptive-field nonlinearities of marmoset retinal ganglion cells. A**: Layouts of receptive fields (shown as 1.5-$\sigma$ contours), spike-train autocorrelograms (lower left), and temporal filters (lower right) of four analyzed cell classes, each showing data from one experiment. The Large OFF cells are pooled from three clusters, each shown separately. Scale bars: spatial: 300 µm, temporal filters: 100 ms, autocorrelograms: 25 ms. **B**: Spatial receptive field (left) and temporal filter (right) of an OFF midget cell (top row) and an OFF parasol cell (bottom row). **C**: Firing rate profiles of the two sample cells from **B** (green trace: OFF midget cell; orange trace: OFF parasol cell) in response to reversing gratings (60 µm spatial period, shown on top; ellipses denoting the 1.5-$\sigma$ receptive-field contours). RGI specifies the reversing-grating index of the corresponding cell. **D**: Distributions of the reversing-grating index for all four cell classes. **E**: Firing rate histograms with equally spaced bins from the training segments for white-noise (WN, top) and naturalistic-movie (NM, bottom) stimulation, based on mean light intensity ($I_{mean}$) and local spatial contrast (LSC) for the OFF midget (left) and OFF parasol (right) cells from **B**. Firing rates are normalized to the histogram maximum, with empty bins shown in gray. **F**: Firing rate histograms with quantile bins computed for regions corresponding to blue and yellow boxes in **E**. Data points used to compute the LSC sensitivity are overlaid, along with the linear fit. Right: Firing rate data (solid lines) of three selected histogram rows (marked by arrows of corresponding color) and fitted softplus functions (dashed lines). **G**: Distribution of LSC sensitivity for all four cell classes under white noise (left) and naturalistic movies (right).

These different classes of ganglion cells have different sensitivity to spatial structure inside their receptive fields, owing to different degrees of nonlinear spatial integration they perform [11,20,23]. For example, compared to the relatively linear midget cells, the more nonlinear parasol cells typically display more pronounced and frequency-doubled responses under contrast-reversing gratings, as we also found in our recordings (Fig 1C). We captured the nonlinearity of each cell's response to such gratings by a reversing-grating index, which was computed from the trial-averaged responses as the ratio of the spectral power at twice the stimulus frequency to that at the stimulus frequency. This index had similar values across cells within each class (Fig 1D) and highlighted differences between midget, parasol, and Large OFF cell clusters.

To understand how these differences manifest themselves for richer stimuli with more spatiotemporal complexity, we analyzed the cells' responses to spatiotemporal white noise as well as to a naturalistic movie. The white-noise stimulus consisted of a flickering checkerboard pattern, while the naturalistic movie comprised segments from a publicly available movie. As visual stimuli that fall onto the retina under natural viewing conditions are also affected by gaze shifts, the naturalistic-movie segments were modified by shifting the movie frames according to a model of gaze shifts (see *Methods*), including saccades and fixational eye movements, consistent with the statistics of marmoset eye movements [24].

To assess how the spatial structure of the stimulus shapes the response of a ganglion cell to such stimuli – beyond the simple, linear activation of the cell's receptive field – we quantified the effective mean light intensity as well as the spatial structure inside the receptive field at each time point of the stimulus. To do so, we first filtered the stimulus by convolving it with the temporal filter of the cell, as obtained from the reverse-correlation analysis, to take care of the cell's temporal stimulus integration.

We then measured the mean light intensity signal, $I_{mean}$, as the weighted average of pixel intensity values (given by their contrast relative to the mean stimulus intensity over time) in the receptive field of the cell. The weights were given by the cell's spatial filter, here defined as a Gaussian fit to the spatial component of the spike-triggered average under white noise. $I_{mean}$ is a measure of the activation of the linear part of the receptive field and is equivalent to the input that is commonly used for computational models that do not account for receptive-field nonlinearities.

We furthermore measured the local spatial contrast, LSC, in each frame of the temporally filtered stimulus. The LSC was computed as the weighted standard deviation of the pixel intensity values within the receptive field, with weights given again by the spatial filter. The LSC provides a measure of the effective spatial stimulus structure that the cell was exposed to at a given point in time.

We analyzed how the cell's activity depended on $I_{mean}$ and LSC by compiling two-dimensional histograms of average firing rates conditioned on different combinations of $I_{mean}$ and LSC, as exemplified in Fig 1E for one OFF midget (left) and one OFF parasol cell (right), considering both the spatiotemporal white noise (top) and the naturalistic movie (bottom). Note that the regions of available combinations of $I_{mean}$ and LSC differ between the two stimuli. Histograms computed from spatiotemporal white noise are distributed around $I_{mean} = 0$ and LSC = 1 with approximate circular symmetry, but those computed from naturalistic movies cover a broader range of $I_{mean}$ values at low LSC and a narrower range of $I_{mean}$ values at high LSC, indicating that the stimulus is dominated by low-spatial-contrast scenes. Furthermore, the range of covered $I_{mean}$ and LSC values under white-noise stimulation can be different for different cells because larger receptive fields, as for the sample parasol cell versus the sample midget cell, lead to more averaging over stimulus pixels and thus smaller ranges of $I_{mean}$ and LSC values.

The histograms reveal that the firing rates of the two cells not only depended on $I_{mean}$, but were also affected by LSC. For example, if LSC had no effect on the firing rate, that is, if stimulus integration in the receptive field were completely linear, regions of high firing rate, corresponding to brighter regions in the plot, should not depend on the value along the *y*-axis and rather be fully vertically oriented. Instead, these regions had a slanted orientation, indicating higher firing rate towards the upper right corner of the plot, where both $I_{mean}$ and LSC are large, in particular for the sample parasol cell. For the sample midget cell, this tendency was less pronounced, suggesting that the LSC had somewhat less influence over the cell's firing.

To quantify the dependence of the firing rate on LSC for each cell and each stimulus condition, we extracted a threshold $I_{mean}$ value for each row in the histogram, that is, for each level of LSC. We did so by fitting a family of parametrized curves to the firing rates of the histogram rows, which had identical shapes for all rows, but allowed for a horizontal shift, whose value defined the threshold value for the corresponding level of LSC (Fig 1F, see *Methods*). We observed that these threshold values, when plotted against the LSC values, generally lay approximately on a straight line. We used the slope of this line, extracted from a linear fit (Fig 1F), to define the LSC sensitivity. The LSC sensitivity is a measure of the relative importance of the LSC to that of the $I_{mean}$ in determining the cell's spiking response, taking on a value of zero when the LSC has no effect and positive values when, for the same $I_{mean}$ value, larger LSC leads to higher firing rate. The LSC sensitivity can thus be seen as a measure of receptive field nonlinearity, equivalent to the reversing-grating index, but assessed under conditions of naturalistic stimulation rather than reversing gratings.

We found that the distribution of the LSC sensitivity is largely consistent across cells within a cell type (Fig 1G). Moreover, cell-type-specific distributions of the LSC sensitivity have relative positions similar to those of the reversing-grating index (Fig 1D). For example, Large OFF cells had the largest values in all conditions, whereas OFF midget cells generally had comparatively small values for both the reversing-grating index and the LSC sensitivity. However, some discrepancies were also apparent, such as the relatively high values of LSC sensitivity under naturalistic movies for ON parasol cells, which were clearly larger than those of OFF parasol cells and nearly on the level of Large OFF cells, suggesting that the ON parasol cells displayed particularly strong receptive-field nonlinearities under naturalistic stimulation.

Another notable discrepancy was in the bimodal distribution of LSC sensitivity values under white-noise stimulation for OFF parasol cells (Fig 1G, left panel), with the two peaks corresponding to OFF parasol cells from different experimental datasets. The reason for this bimodality is unclear. While this might reflect experimental variability, differences in retinal locations or eccentricity of the extracted tissue might have contributed here. Eccentricity, for example, can influence receptive-field size [25], amacrine cell density [26], and cone photoreceptor density [27], though across the cells of our two datasets, we did not observe significant differences in receptive-field sizes (118 μm ± 12 μm versus 115 μm ± 9 μm, mean ± standard deviation, for OFF parasol cells, the only cell type consistently observed in both datasets; Welch's t-test, $p = 0.26$, $n_1 = 31$ and $n_2 = 38$).

To illustrate the relation between LSC and firing rate, Fig 2 shows examples of the $I_{mean}$ and LSC computation together with firing-rate responses from a sample OFF parasol cell (Fig 2A) and a sample OFF midget cell (Fig 2B) for different segments of the naturalistic movie. Each row displays the computation of $I_{mean}$ and LSC at a given point in time from the corresponding temporally convolved movie frame, which here resulted in nearly identical $I_{mean}$ signals (4th column) for each sample cell, but substantially different LSC signals (5th column).

Comparing the temporal progression of the $I_{mean}$ and LSC signals over time (Fig 2, 6th column, top plots in each row) with the corresponding firing rates (Fig 2, 6th column, bottom plots in each row) shows that the larger LSC values yielded much larger firing rates at the time points of interest (light gray regions), compared to the scenarios with smaller LSC values. This is true even for the OFF midget cell, despite the typically more linear receptive field of this cell type, and follows from the substantial difference in LSC in the example. For the OFF parasol cell, note that the earlier firing-rate peak visible in the top row of Fig 2A corresponds to an earlier stimulus segment and thus to different $I_{mean}$ and LSC values, in particular a higher $I_{mean}$ signal as seen in the corresponding trace. These examples illustrate how local spatial contrast can shape the firing response of ganglion cells, corroborating that the LSC can be used to capture these effects in computational models.

## Predictive model with spatial-contrast sensitivity

In order to incorporate the LSC into a predictive model for responses to spatiotemporal stimuli, we extended the spatial contrast (SC) model that had previously been used to predict responses to flashed natural images in the salamander and mouse retina [21]. The SC model itself is an extension of the commonly used linear–nonlinear (LN) model [2]. In the LN model, the first, linear part of the model filters the stimulus with the receptive field of the cell to yield the weighted mean of

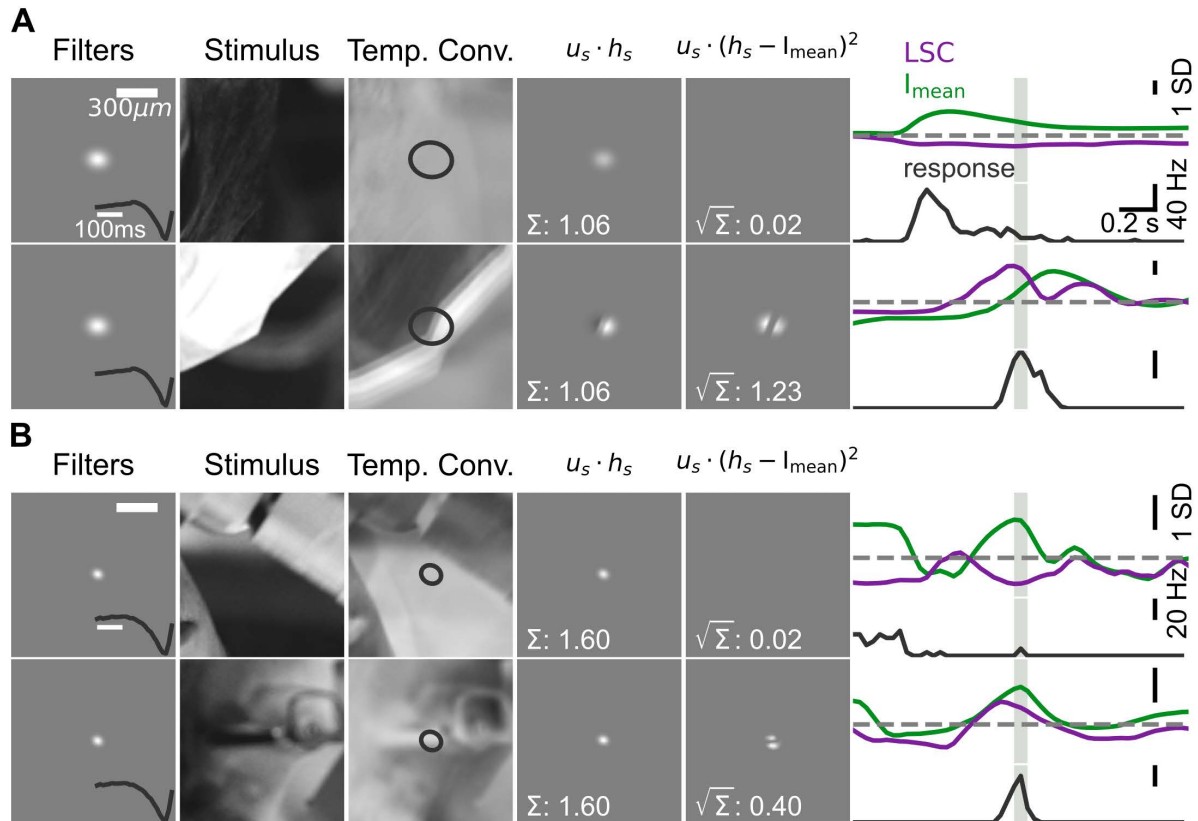

**Fig 2. Examples of the effect of local spatial contrast on responses to natural scenes. A**: Illustration of the stimulus processing to obtain $I_{mean}$ and LSC by applying the filters of a sample OFF parasol cell (1st column) to the stimulus (2nd column, exemplified here by a single frame) to obtain a temporally convolved stimulus (3rd column, overlaid with the $3\text{-}\sigma$ contour of the spatial filter), from which the weighted light intensity signal (4th column) and the weighted spatial contrast (5th column) are extracted. The two rows correspond to two separate stimulus segments. The plots of the light-intensity and spatial-contrast signals display the results of the pixel-wise computation as indicated by the formulas at the top (where $u_s$ denotes the spatial filter and $h_s$ the temporally convolved stimulus frame), with the final values given in the lower left of the plots. The time course of the Z-scored light-intensity and spatial-contrast signals (top) along with the corresponding measured firing rates (bottom) of the sample cell are shown in the 6th column, with the light gray region highlighting the time window of presentation of the analyzed stimulus frame for which the $I_{mean}$ and LSC values in columns 4 and 5 were calculated. **B**: Same as **A**, but for a sample OFF midget cell. Both cells are the same as in Fig 1B–F. Frames of the stimulus taken from an open-source movie licensed under CC BY 3.0 [28].

the light intensity in the stimulus, with weights given by the receptive field, equivalent to computing the cell's $I_{mean}$ signal. The nonlinear part of the LN model then transforms the weighted mean light intensity into the model's response (e.g., firing rate or spike probability) by applying a nonlinear function, the output nonlinearity.

In the SC model, in addition to the $I_{mean}$ signal, the LSC is measured for the stimulus, and the two are linearly combined, using a weight parameter $w$, which determines the importance of the local spatial contrast relative to the mean light intensity signal. This enables the SC model to scale the dependence on LSC independently for each cell. The linearly combined signal is then transformed into the model's response by an output nonlinearity, much like in the LN model. For both models, we here applied a softplus rectifying nonlinearity that is parameterized with three free parameters.

In our spatiotemporal version of the SC model, the calculation of $I_{mean}$ and LSC is preceded by a temporal convolution of the stimulus with the cell's temporal filter. The full model is summarized in the schematic of Fig 3. Together, implementing the SC model for a given cell requires estimating the cell's spatial receptive field and temporal filter, which can both be obtained by reverse correlation under spatiotemporal white-noise stimulation, as well as optimizing four free parameters

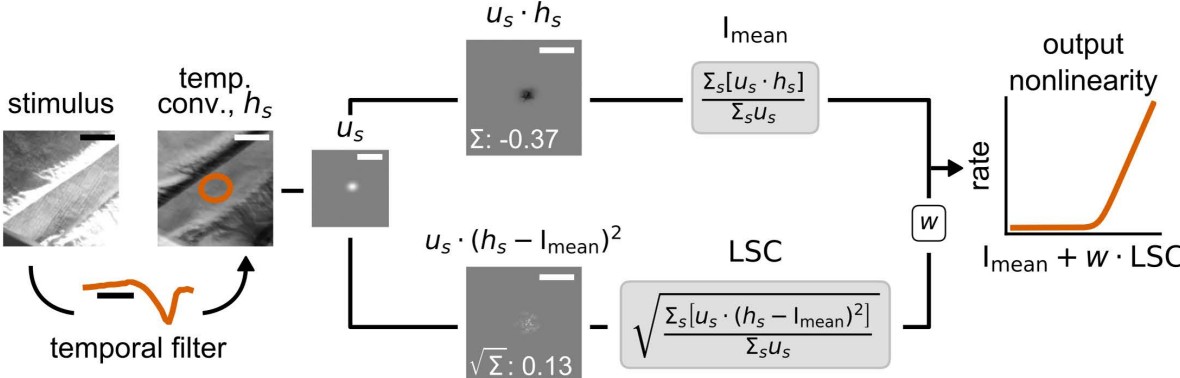

**Fig 3. Schematic of the spatial contrast (SC) model.** For a given cell, the incoming stimulus is first convolved with the cell's temporal filter to yield the effective stimulus frame $h_s$ (overlaid here with the 3-$\sigma$ contour of the spatial filter). From this, the mean light intensity signal $I_{mean}$ and the spatial contrast signal LSC are computed by using the cell's spatial filter $u_s$ as weights. The $I_{mean}$ and LSC are linearly combined with a weight parameter $w$ used to scale the LSC contribution, and the combined signal is transformed by an output nonlinearity to yield the model's firing rate response. Scale bars: spatial: 300 µm, temporal: 100 ms. Frame of the stimulus taken from an open-source movie licensed under CC BY 3.0 [28].

– three for the softplus nonlinearity and one for the weight $w$ of the LSC. We here optimized the free parameters simultaneously, by minimizing the negative log-likelihood under the assumption of a Poisson spiking process.

## Cell-type-specific comparison of LN and SC models

We next tested whether the SC model improves model predictions as compared to the LN model for white noise as well as for naturalistic movies. Both stimuli were presented to the retina in several trials, each consisting of a unique non-repeating segment used for model training and a fixed stimulus segment, repeated in every trial, which was used for model testing. For both models, we applied the same spatial filter, which was here obtained as a two-dimensional Gaussian fit to the spatial receptive field as extracted from the spike-triggered average. In principle, however, the filter could also come from any other available method of receptive-field estimation, such as responses to flashed spots of light, thus keeping the approach simple and versatile. Finally, model performances were evaluated for each cell on the time-dependent firing rate averaged across trials of the test segments by computing the Pearson correlation with the model prediction (Fig 4A).

For spatiotemporal white noise, we found that the SC model yielded better model predictions than the LN model across all cell classes (Fig 4B, top row). The improvement was particularly notable for the ON parasol cells (average ratio of correlation values $r_{SC}/r_{LN}$ ± standard deviation: 1.17 ± 0.03, $p < 0.0001$, one-sided Wilcoxon signed-rank test) and Large OFF cells ($r_{SC}/r_{LN}$: 1.41 ± 0.32, $p < 0.0002$), whereas OFF midget cells displayed a more modest improvement ($r_{SC}/r_{LN}$: 1.09 ± 0.05, $p < 0.0001$). OFF parasol cells exhibited greater variability in model improvement and a smaller average improvement compared to OFF midget cells ($r_{SC}/r_{LN}$: 1.08 ± 0.08, $p < 0.0001$), owing to the discrepancy of this cell type across the two analyzed datasets. On splitting the population of OFF parasol cells by dataset, we found that, while both populations were significantly better predicted by the SC model than by the LN model ($r_{SC}/r_{LN}$: 1.16 ± 0.08, $p < 0.0001$, versus 1.02 ± 0.01, $p < 0.0001$), the cells in one dataset were distinctly more nonlinear than in the other. This aligns with our observation regarding the distribution of LSC sensitivity values for OFF parasol cells under white-noise stimulation (Fig 1G).

For naturalistic movies (Fig 4B, bottom row), the performance gain by including LSC information in the model was generally lower than for white noise, yet the SC model still outperformed the LN model for ON parasol cells ($r_{SC}/r_{LN}$: 1.06 ± 0.03, $p < 0.0001$), OFF parasol cells ($r_{SC}/r_{LN}$: 1.05 ± 0.07, $p < 0.0001$) and Large OFF cells ($r_{SC}/r_{LN}$: 1.32 ± 0.13, $p < 0.0001$). However, no change in the prediction performance was observed for OFF midget cells ($r_{SC}/r_{LN}$: 1.00 ± 0.02, $p = 0.83$).

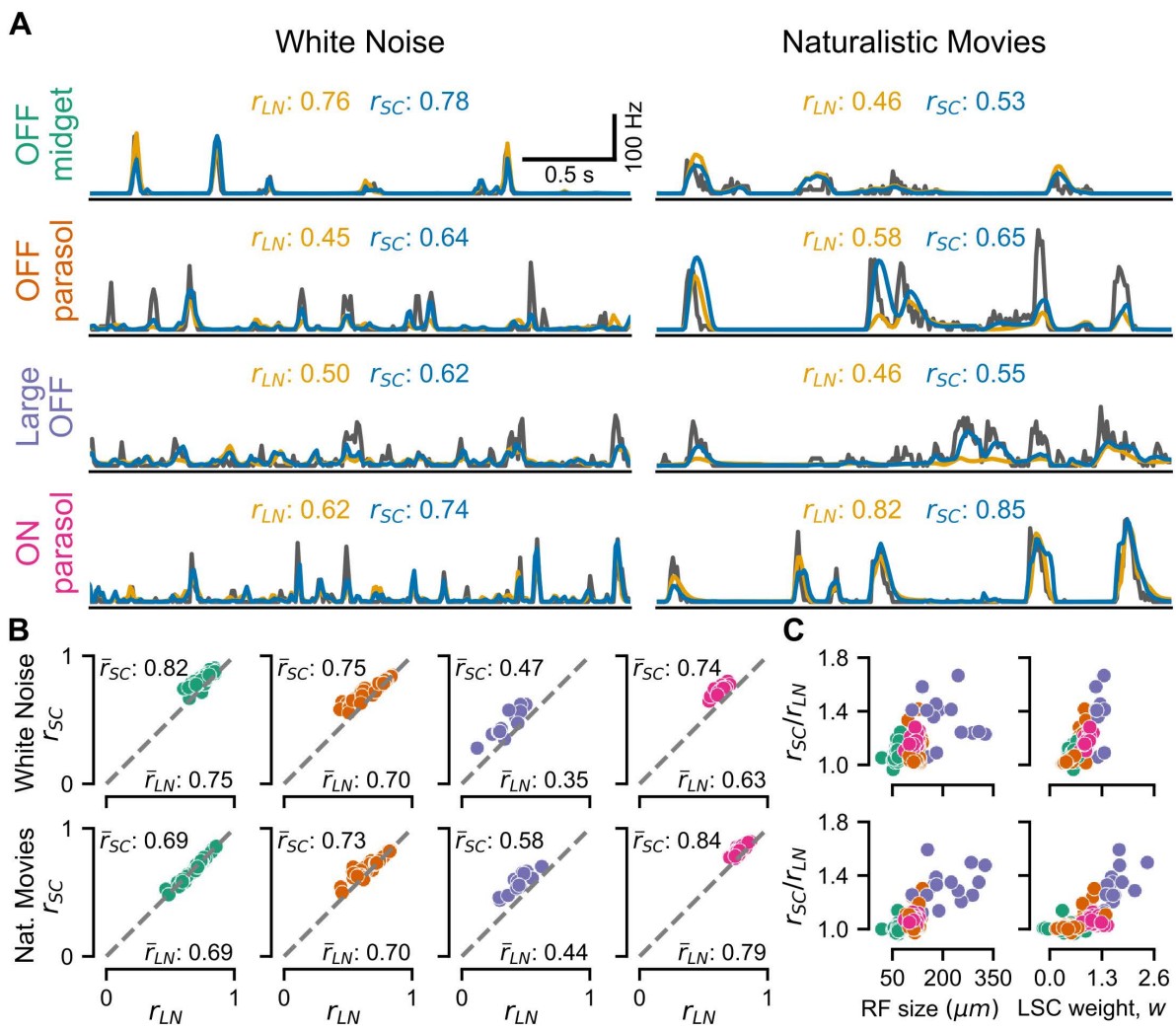

**Fig 4. Evaluation of model performance by cell type. A**: Responses of sample cells from each of the four cell classes in Fig 1A to 3 s segments of test stimuli (gray) for white noise (left column) and naturalistic movie (right column), together with the predictions from the LN model (orange) and the SC model (blue). The numbers above each plot show the Pearson's correlation coefficients between model prediction and cell response. **B**: Comparison of correlation coefficients for the SC model ($r_{SC}$) and the LN model ($r_{LN}$) for each cell in the four cell classes (colors corresponding to the labels in **A**) under spatiotemporal white noise (top row) and naturalistic movie (bottom row). The average correlation coefficients $\bar{r}_{SC}$ and $\bar{r}_{LN}$ are shown next to the corresponding axes. **C**: Relative improvement in model performance ($r_{SC}/r_{LN}$) under white noise (top row) and naturalistic movies (bottom row) for each cell versus receptive-field size (left) and the obtained LSC weight $w$ (right). Colors indicate cell class as in **B**.

The finding that models of OFF midget cells profit little from including spatial contrast information is consistent with their weak responses to reversing gratings (Fig 1D) and the smaller dependence on spatial contrast in firing rate histograms (Fig 1E) as well as with the notion that midget cells are classically considered to have rather linear receptive field [29]. Note, though, that more recent reports have shown that midget cells also display substantial receptive-field nonlinearities, for example, as demonstrated in the peripheral macaque retina [15,20].

The results suggest that cells with larger receptive fields have a greater improvement in model performance by including spatial contrast information. This is confirmed by plotting the ratio of model performances $r_{SC}/r_{LN}$ against receptive-field size (Fig 4C, left). Cell types with larger receptive fields display greater relative improvement in model

performance. For midget cells under white-noise stimulation, this trend is also apparent for cells of the same type (Pearson's $r$: 0.41, $p = 0.003$, two-tailed t-test). That larger cells are more sensitive to spatial contrast makes intuitive sense, as their larger receptive fields are more likely to actually experience substantial spatial structure. For OFF midget cells under natural stimuli, for example, there may simply be too little spatial structure available at the high spatial frequencies that would fit into their small receptive fields. Thus, unlike for white-noise stimuli where signal power extends to high frequencies, midget cells might simply not experience sufficient spatial contrast to shape their responses beyond what comes from the activation of the linear receptive field.

We also checked how the improvement in model performance is related to the weight parameter $w$, which sets the relative contribution of the spatial contrast information in determining the activation of the SC model. As expected, we found that larger relative improvements of the SC model over the LN model were tightly associated with larger weights (Fig 4C, right). A weight of zero reduces the SC model to an LN model, allowing no improvement, and larger positive weights reflect a stronger relevance of LSC on the cell's activity and thus a more nonlinear receptive field.

To illustrate how the SC model can lead to improvements in model prediction over the LN model, we examined the responses and model firing rates of a sample OFF parasol cell for different episodes taken from the test data of the naturalistic movie (Fig 5A). In three of the four representative cases (Fig 5B, first three rows), the LN model predicted a firing response lower than the ground truth, owing to the low average light intensity in the receptive field. Here, the temporally convolved stimulus, weighted by the cell's receptive field (Fig 5B, 3rd column) displayed bright as well as dark regions of similar extent, not unlike a grating stimulus. The substantial spatial structure inside the receptive field led to large values of the local spatial contrast, which in turn allowed the SC model to predict higher firing rates in response to the displayed stimulus episodes, which better matched the response of the cell.

Moreover, the additional spatial contrast information allowed the SC model to also predict a lower firing response than the LN model. This is due to the fact that, at the same mean light intensity, the SC model can predict different responses to different local spatial contrast values in the scene, whereas the LN model predicts the same, average response across the different values of local spatial contrast. The final example case (Fig 5B, last row) illustrates this for one episode where the lack of spatial structure within the receptive field suppressed the prediction of the SC model, which was closer to the ground truth than the higher prediction of the LN model.

## Comparison of model performance with a subunit model

So far, we have shown that the SC model makes use of local spatial contrast to yield better response predictions than the LN model, suggesting that most cells are sensitive to high-frequency spatial contrast. In previous work, a standard modeling approach to capture this sensitivity has been to apply subunit models [10–13,19,30], which build up a ganglion cell's receptive field from a set of smaller subunits, whose signals are nonlinearly combined. Compared to the SC model, obtaining the subunits is considerably more data-intensive. Yet, comparing the performance in predicting responses to stimuli with fine spatial structure can serve to evaluate the power and limits of the SC model.

We therefore constructed subunit models from the white-noise responses of our recorded cells and compared the prediction performance to that of the SC model. To infer the spatial filters corresponding to functional subunits, we performed spike-triggered covariance (STC) analysis [31–33] for the temporally-convolved white-noise stimuli, which yielded a set of orthogonal spatial filters (Fig 6A). As proposed previously [16], we then transformed the set of filters according to a probabilistic spiking model that combined the filter signals akin to a logical-OR operation (see *Methods*). This yielded a new set of filters, which resembled spatially localized subunits (Fig 6B) and therefore formed the basis of our subunit model. Since the eigenvalue spectrum of the STC analysis (Fig 6C) typically did not clearly specify the dimensionality of the relevant stimulus space and thus the number of required filters, the number of subunits for each cell (Fig 6D) together with the subunit weights was determined by optimization of the subunit model using training and held-out validation data. In the model, the subunit signals were half-wave rectified and their weighted sum was passed through a final parameterized nonlinear

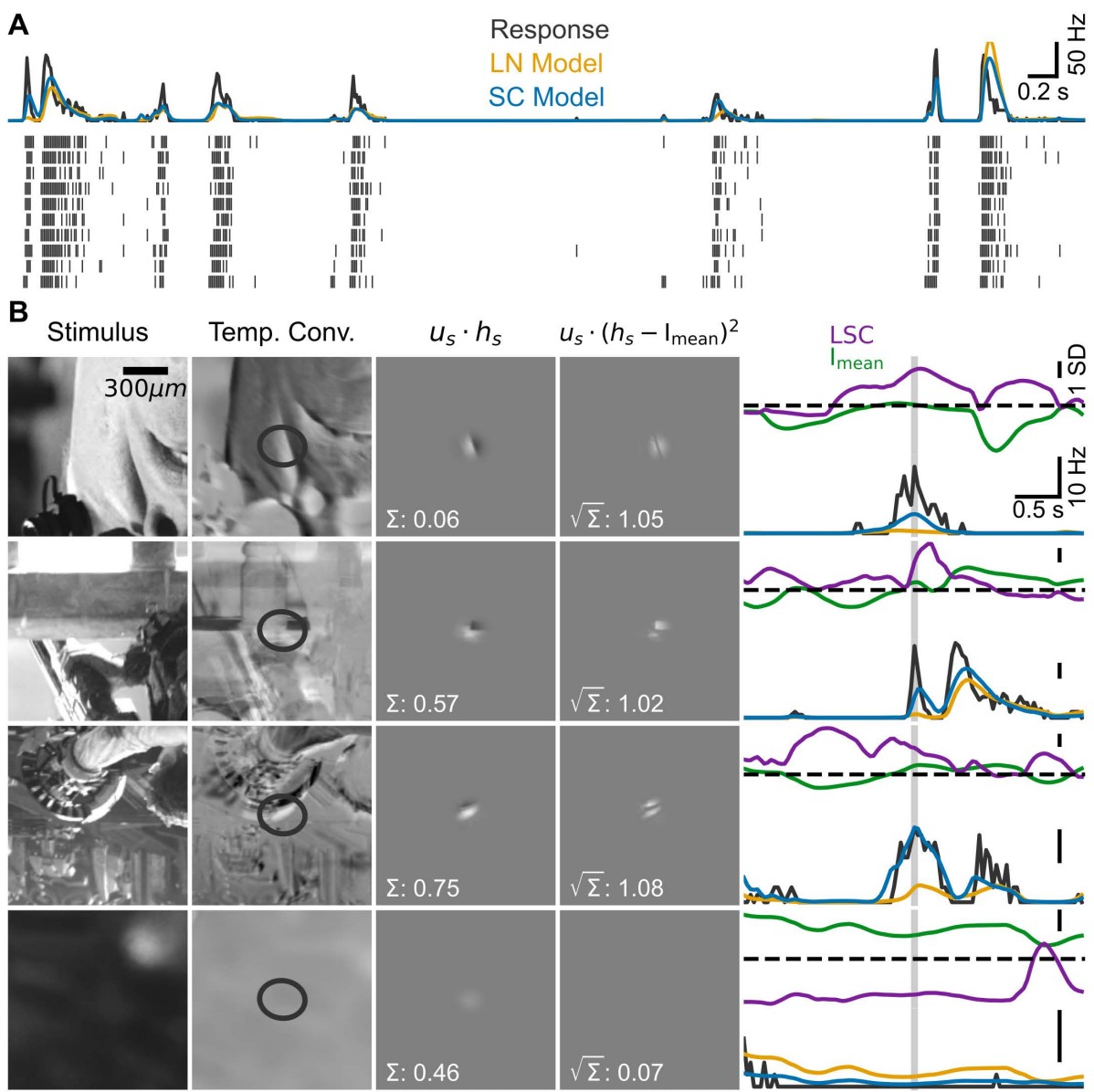

**Fig 5. Examples of improved predictions by SC model versus LN model. A**: Firing rate (top, dark gray) and spike raster (bottom) for the sample OFF parasol from Fig 1B to an 8 s segment of the naturalistic movie test stimulus. Overlaid are response predictions for the LN model (orange) and the SC model (blue). **B**: Examples of episodes from the natural movie test stimulus with substantially better response prediction by the SC model versus the LN model for the same sample cell as in **A**. The columns are shown in the same way as in Fig 2. The firing rate curves in column 5 are overlaid with response predictions with colors as in **A**. Frames of the stimulus in **B** taken from an open-source movie licensed under CC BY 3.0 [28].

function to yield the model prediction, which was compared to the measured response and the prediction of the SC model on the test data (Fig 6E).

For most RGCs in our datasets, we found that the SC model could predict the cells' responses to spatiotemporal white noise better than the subunit model (Fig 6F). The improvement was most pronounced for Large OFF cells (average ratio of correlation values $r_{SC}/r_{sub}$ ± standard deviation: 2.95 ± 0.66, $p < 0.0001$, one-sided Wilcoxon signed-rank test), with

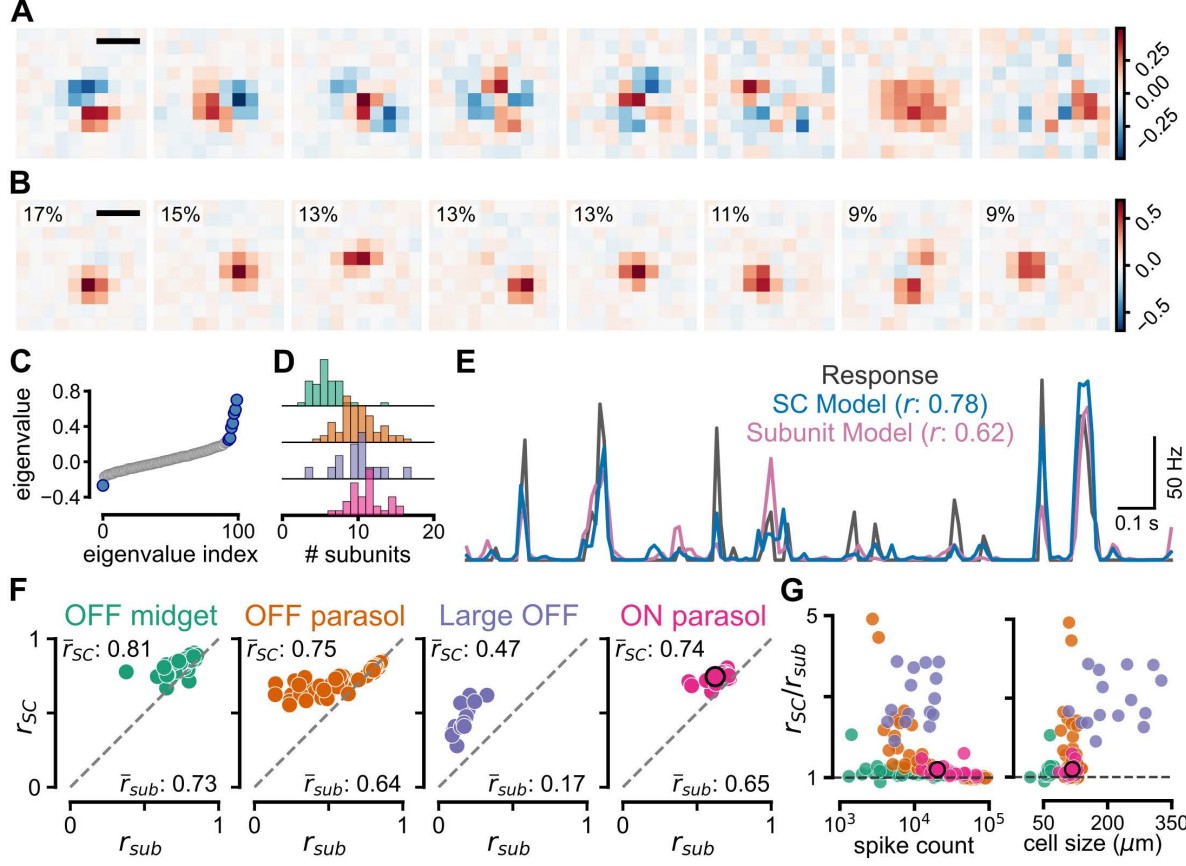

**Fig 6. Computation of functional subunits and performance comparison with a subunit model. A**: Filters representing the relevant stimulus subspace computed via spike-triggered covariance (STC) for a sample ON parasol cell. Shown here are the top 8 of the 16 used for modeling. Scale bar: 100 μm. **B**: Inferred filters of the functional subunits for the sample cell. The optimal number of filters (here 8) was chosen during model training. The relative weight of each filter in the subunit model is displayed at the top-left of each panel as a percentage of the total sum across all filters. Scale bar: 100 μm. **C**: Eigenvalue spectrum from the STC analysis of the sample cell. Blue circles correspond to filters from **A**. **D**: Distribution of the obtained number of subunits for the four cell classes. Colors as in **F**. **E**: Firing rate of the sample cell (dark gray) overlaid with predictions from the SC model (blue) and the subunit model (pink) for a 2 s segment of the spatiotemporal white-noise stimulus. Pearson's correlation coefficient between model prediction and cell response displayed next to model name in legend. **F**: Comparison of correlation coefficients for the SC model ($y$-axis, $r_{SC}$) and the subunit model ($x$-axis, $r_{sub}$) for each cell of the four cell classes under spatiotemporal white noise. The average correlation coefficients $\bar{r}_{SC}$ and $\bar{r}_{sub}$ are shown next to the corresponding axes. **G**: Relative improvement in model performance ($r_{SC}/r_{sub}$) under white noise for each cell versus the number of spikes in the training segment of the stimulus *(left)* and the size of the cells' spatial receptive fields *(right)*. Colors as in **F**. In panels **F** and **G**, the sample ON parasol cell from **A**–**E** is highlighted with a black circle.

modest improvements seen also for OFF midget cells ($r_{SC}/r_{sub}$: 1.13 ± 0.15, $p < 0.0001$), OFF parasol cells ($r_{SC}/r_{sub}$: 1.35 ± 0.72, $p < 0.0002$), and ON parasol cells ($r_{SC}/r_{sub}$: 1.14 ± 0.12, $p < 0.0001$).

While it may seem surprising that the low-parametric SC model outperforms the more complex subunit model, the fewer tunable parameters lend the SC model a distinct advantage when faced with limited experimental data. For example, the ratio of the correlation coefficients, $r_{SC}/r_{sub}$, for an OFF parasol or an ON parasol cell was comparatively smaller, corresponding to a relatively better subunit model performance, when more spikes were available in the training segment of spatiotemporal white noise ([Fig 6G](link), left). Moreover, subunit models also tended to perform closer to the SC model for cell types with smaller receptive fields ([Fig 6G](link), right), which require fewer parameters in the spatial filters to cover the receptive-field area. Thus, the average performance of the subunit model would likely increase if larger datasets were

available or if additional constraints like regularization or parameterization of subunits were included in the analysis. Nonetheless, the absence of a clear advantage in model performance by including subunits even for the cells with the largest amount of training data suggests that the SC model presents a competitive alternative for phenomenologically capturing nonlinear spatial integration. Furthermore, the analysis illustrates the robustness and computational simplicity that come with the SC model.

**Identifying spatial scales of stimulus integration with the SC model**

Visual input can contain information at a wide range of spatial frequencies, but not all of these are relevant for downstream sensory processing. Spatial contrast at very high spatial frequencies, corresponding to scales below the size of bipolar cell or even photoreceptor receptive fields, for example, is unlikely to matter even for a nonlinear ganglion cell. One may thus ask what the relevant spatial scales of integrating spatial contrast are for the analyzed marmoset retinal ganglion cells. This question can be targeted with the SC model as a tool by analyzing what the optimal spatial scale is for assessing local spatial contrast so that only contrast above this spatial scale contributes to the model's activation.

To do so, we used a modified version of the SC model, in which the LSC is computed from the temporally convolved stimulus after additional spatial smoothing, here performed with a two-dimensional circular Gaussian filter ([Fig 7A]). The smoothing eliminates spatial-contrast information on scales smaller than the spatial scale of the Gaussian filter but retains spatial-contrast information on larger scales. The smoothing thus effectively implements a (soft) cutoff for high spatial frequencies, similar to how a subunit model would be insensitive to spatial structure finer than the individual subunits. The computation of $I_{mean}$, on the other hand, is kept the same as in the original model.

For each cell, we thus trained the modified SC model for various spatial scales of smoothing (defined as three times the standard deviation of the Gaussian filter, in line with our definition of the size of a cell's receptive field). The resulting performance in predicting responses to the test stimulus was then compared with that of the original SC model without stimulus smoothing ([Fig 7B]). For spatiotemporal white noise, a clear pattern emerged. Across all cell types, increasing the scale of spatial smoothing led, on average, to an initial increase in performance followed by a steady decrease ([Fig 7C], top row). The relative performance peaked at a spatial scale of around 40–50 μm (distribution of peaks across cells shown in insets of [Fig 7C], top row; mean ± standard deviation in units of μm: all cells: 48 ± 19; OFF midget cells: 45 ± 33, OFF parasol cells: 51 ± 4, Large OFF cells: 47 ± 4, ON parasol cells: 46 ± 2).

The magnitude of the model improvement obtained by assessing the local spatial contrast with the optimal scale of smoothing can be read off from the peak size of the performance-improvement curves. This shows that the improvement of SC model performance by adjusting the spatial scale can be considerable, up to around 10% for individual ON and OFF parasol cells and even 20% for individual Large OFF cells. Over the population of all cells, the improvement was consistently substantial for OFF parasol cells (relative Pearson's $r$: 1.05 ± 0.04, mean ± standard deviation, $p < 0.0001$, one-sided Wilcoxon signed-rank test), ON parasol cells (relative Pearson's $r$: 1.06 ± 0.02, $p < 0.0001$), and Large OFF cells (relative Pearson's $r$: 1.11 ± 0.05, $p < 0.0001$), and less so (though still significant) for OFF midget cells (relative Pearson's $r$: 1.02 ± 0.02, $p < 0.0001$). This likely reflects the more nonlinear receptive fields of the parasol and Large OFF cells, whose models can therefore benefit more from assessing spatial contrast information at the right scales.

For the naturalistic movie stimulus, the same analysis of different smoothing scales did not reveal optimal scales for any of the four cell classes ([Fig 7C], bottom row). Increasing the scale of spatial smoothing overall had little effect on model performance and did not systematically improve model performance for any of the cell types. Note, though, that this does not mean that including spatial contrast information would not be beneficial for response predictions under natural stimuli; the baseline in [Fig 7C] is from a spatial contrast model with no smoothing, that is, no threshold at high spatial frequencies. The flat curves simply mean that smoothing in order to cut irrelevant spatial contrast has no effect.

The stark difference between the two stimuli in the effect of stimulus smoothing can be explained by the fact that natural stimuli are dominated by low spatial frequencies. To illustrate this, we computed the radially-averaged power spectral

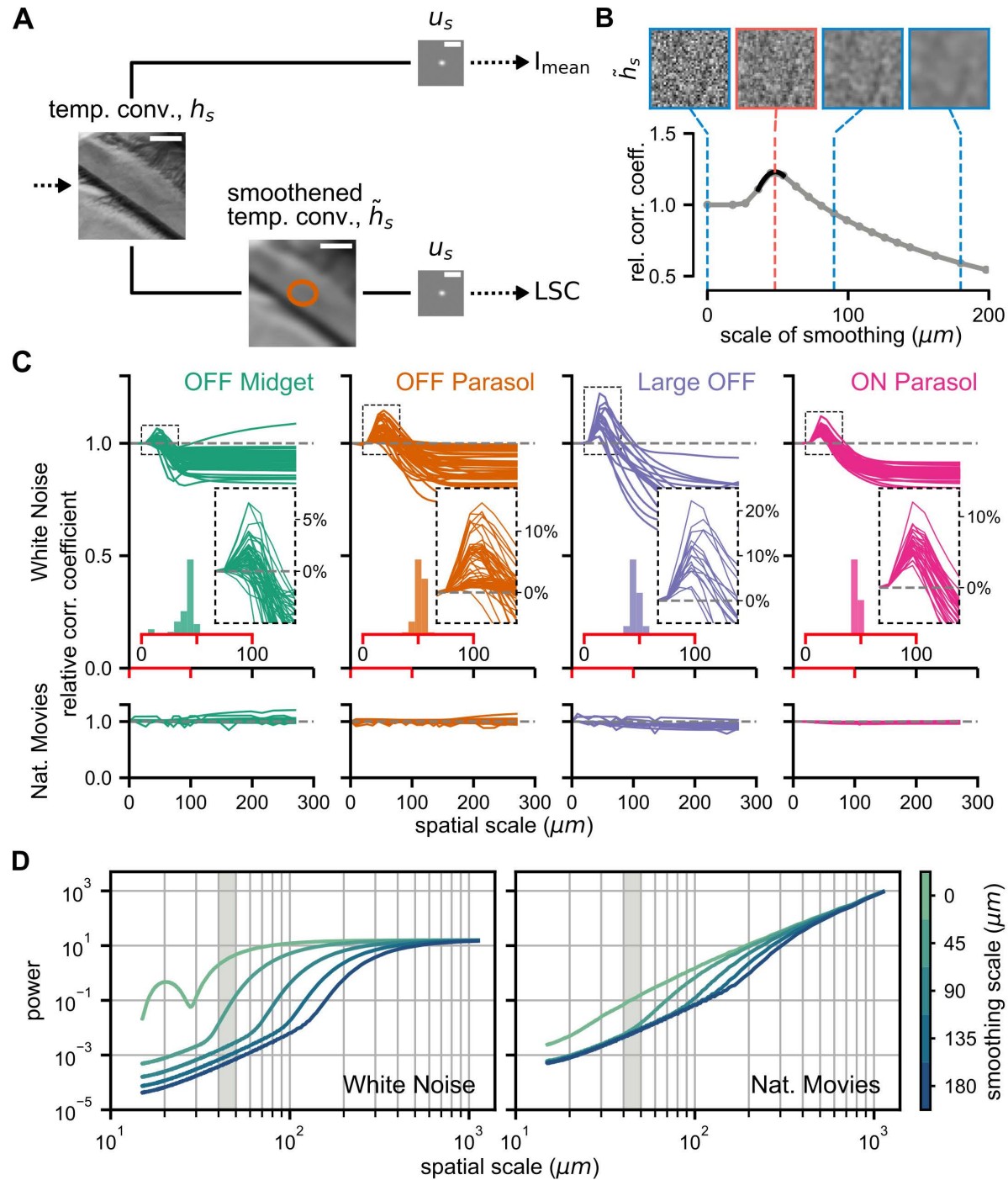

**Fig 7. Analysis of spatial scale of contrast sensitivity. A**: Schematic of the SC model from Fig 2, modified to include spatial stimulus smoothing before the computation of LSC. **B**: Spatial smoothing analysis for a sample cell in response to white noise. Gray curve shows the SC model performance for different smoothing scales, normalized to performance without smoothing. The optimal scale is obtained by interpolation (black segment). Sample white-noise frames at different smoothing scales are shown on top, with the red outline marking the approximate optimal scale. **C**: Spatial smoothing analysis for all cells under white-noise (top) and naturalistic-movie (bottom) stimulation, with the region around the peak shown enlarged in the inset with dashed outline and relative model performance shown as percentage improvements over a model without smoothing. Distributions of spatial scales of nonlinear stimulus integration for each cell class under white noise are shown in the inset histograms. The *x*-axis here corresponds to the range of the red segment of main plot. **D**: Radially-averaged power spectral density (RAPSD) curves averaged across frames of white noise (left)

and of the naturalistic movie (right) for different scales of spatial smoothing. Spatial scale (*x*-axis) is here defined as the inverse of spatial frequency. Shaded gray areas mark the range of 40–50 μm, matching approximately the range of spatial scales from **C**. Note that the RAPSD of the non-smoothed white-noise stimulus falls off for small spatial scales because RAPSD curves were here computed based on monitor-pixel resolution and the stimulus was close to white only for scales beyond the length of two stimulus squares. Frame of the stimulus in **A** taken from an open-source movie licensed under CC BY 3.0 [28].

density (RAPSD) for each stimulus and compared the spectra at different scales of spatial smoothing (Fig 7D). For the white-noise stimulus, the spectrum of the non-smoothed stimulus is flat, as expected, for spatial scales that are larger than twice the size of the stimulus squares (30 μm), and smoothing has a strong effect on the power in the range of several tens of micrometers, in particular near the spatial scales of around 40–50 μm extracted from Fig 7C, thus substantially affecting the computation of LSC.

For the naturalistic movie, on the other hand, the RAPSD curve falls off strongly towards smaller spatial scales even without smoothing, owing to the dominance of low spatial frequencies in natural scenes [34,35]. Thus, the LSC is largely determined by spatial structure at larger scales and smoothing has little effect on the LSC or may even deteriorate the LSC signal, as it diminishes relevant spatial structure at scales beyond the spatial scales of nonlinear stimulus integration of the cells.

## Discussion

Despite the ubiquity of spatial nonlinearities within the receptive fields of retinal ganglion cells [8], current models of retinal stimulus encoding still often start with a linear receptive field, implemented by a spatial filter that is applied to the incoming stimulus. The widespread use (and success) of these linear-receptive-field models is undoubtedly related to the relative ease with which their parameters, in particular the filters themselves, can be obtained from experimental data. Under Gaussian white-noise stimulation, for example, the spike-triggered average can provide an accurate and easy-to-obtain estimate of the filter [2,36]. In addition, natural stimuli allow obtaining the filter via maximum-likelihood procedures [37–39], backed by a good mathematical understanding of the parameter optimization challenge, or via information-theoretic tools [40]. This has supported the success of simple single-filter-based models, such as the LN model and its various extensions, which add model components that capture, for example, cell-intrinsic feedback [41,42], dependencies on other neurons [37], biophysical adaptation models [43], and spike-timing dynamics [44,45].

Extending the models to include receptive-field nonlinearities, on the other hand, has presented a substantial challenge. A typical approach is to split up the receptive field into spatial subunits, which act as multiple, parallel filters and whose signals are nonlinearly combined, often by applying a threshold-linear rectification to the individual subunit signals before additive pooling [8,11–13,19,30,46,47]. Yet, although such subunit models of retinal ganglion cells have been introduced almost half a century ago [10], it has remained difficult to optimize subunit models with respect to data from actual, experimentally measured cells. Among the problems here are the fact that the optimal number of subunits is not known beforehand, which makes parameterizing subunit models difficult, and that the small spatial scale of subunits requires finely structured visual stimuli to resolve their boundaries, which often results in inefficient stimulation and the need for long recording durations.

We therefore turned to a simpler, phenomenological approach for including effects of nonlinear spatial integration into computational models, aiming to keep the advantage of conceptual model simplicity and relatively easy parameter optimization as in linear-receptive-field models. To do so, we consider the local spatial contrast (LSC) inside a cell's receptive field, which depends on the squared deviations of pixel intensities from the mean intensity. Including LSC information can thereby capture spatial nonlinearities of the receptive field. A square-wave spatial grating that symmetrically covers the receptive field with its bright and dark regions, for example, yields zero activation of the linear receptive field but a high LSC signal.

The choice of the weighted pixel variance as a measure to extend the LN model is motivated by its simplicity, by the fact that it directly captures spatial structure beyond the mean intensity, and by its computational independence from the mean-intensity signal. Akin to a Taylor series, the addition of the LSC can be viewed as the natural first-order extension of the linear receptive field. Other single-measure extensions might be feasible, e.g., pixel-intensity entropy or a quantification of spatial autocorrelation, but this would likely yield a measure that is highly correlated with the LSC or abandon the independence from the mean-intensity signal and the intuitive simplicity.

We here first demonstrated how the LSC can be used to measure the level of spatial nonlinearity and sensitivity to spatial contrast in ganglion cells of the primate retina (Fig 1 and Fig 2). We then extended the spatial contrast (SC) model previously suggested for flashed images [21] to spatiotemporal stimuli by accounting for temporal integration before assessing the LSC (Fig 3) and applying a modified measure of LSC that is more appropriate for the variations in receptive-field sizes as encountered, for example, for the cell types of the primate retina. Evaluating on data that we recorded from the marmoset retina, we found that, compared to an LN model with a linear receptive field, the SC model yielded improved response predictions under artificial as well as naturalistic stimulation in a cell-type-specific fashion (Fig 4), owing to its ability to capture nonlinear spatial integration in the receptive field of the cell (Fig 5). Furthermore, we compared the SC model to a subunit model fitted via spike-triggered covariance analysis and found that, despite its simpler structure, the SC model could match or outperform the subunit model (Fig 6). Finally, we demonstrated that the SC model can be employed to identify the spatial scale beyond which nonlinear stimulus integration takes place in the primate retina by comparing model performance with different levels of spatial smoothing prior to the computation of the local spatial contrast (Fig 7).

## Advantages and limitations of the SC model

An important feature of the SC model, in particular in comparison to subunit models, is that matching the model to a given recorded cell mostly requires an estimate of the receptive field and temporal filter, which can be obtained from standard stimuli and analyses, such as reverse correlation under spatiotemporal white-noise stimulation. Beyond this, the model has few free parameters – the weight factor of the local spatial contrast along with the parameters that describe the nonlinearity. Thus, as receptive field measurements are among the typical standard characterizations that form part of any recording in sensory systems, obtaining the SC model essentially amounts to fitting a handful of parameters, which is feasible even with little training data.

Obtaining the receptive field and the temporal filter from white-noise stimulation, however, disregards that adaptation to different stimulus statistics, such as those of natural stimuli, may change a cell's filtering characteristics [48,49]. An interesting future extension might thus be to directly fit the full model, including its spatial and temporal filter, under the stimulus statistics of interest and compare with the filtering characteristics under white noise. It may even be feasible to allow for different spatial filters for computing the mean light intensity signal and the local spatial contrast. Note, though, that fitting the full model substantially complicates the computational demands of the approach because appropriate filter parameterizations and perhaps regularization are required and because $I_{mean}$ and LSC would need to be recomputed repeatedly for each time point during the model optimization process. Moreover, separate filters for $I_{mean}$ and LSC would complicate the model further, as the computation of LSC couples the two measures through the inherent subtraction of $I_{mean}$. In the present work, however, our goal has been to provide a simple modeling framework that can be applied even if the modeled data would not allow filter estimation and if receptive fields are instead obtained independently, e.g., from recordings to flashed spots of light. Furthermore, from a biological perspective it seems plausible that evaluation of light intensity and spatial contrast occurs with the same spatial weighting, as the primary basis for both is thought to be the pooling of excitatory inputs over the presynaptic bipolar cells [17,30,50].

Another current simplification is the assumption that spatial and temporal filtering are separable. This does not only reduce the number of parameters required for describing the filters (and therefore makes the parameter estimation more

robust to noise) but also conceptually simplifies the assessment of the local spatial contrast. Without assuming space-time separability, for example, it is not immediately clear how the spatiotemporal weighting in the computation of the standard deviation should occur, as temporal variations in illumination most likely contribute differently (or not at all) to nonlinear signal processing as compared to spatial structure. Similarly, the current approach neglects the receptive-field surround, which may – besides standard (linear) surround suppression – also contribute nonlinear components [51] or influence the nonlinearities of the receptive field center [52]. An interesting extension could thus be to apply the SC model in the context of a difference-of-Gaussians receptive field [53], with local spatial contrast independently evaluated in the center and in the surround Gaussian. This may, however, require a different estimation of the receptive field, as spatiotemporal white noise often does not reveal proper receptive-field surrounds [54], consistent with the present data (cf. Fig 1B).

## Alternative approaches

Despite the difficulties associated with subunit models discussed above, a number of approaches have successfully tackled the challenge of identifying subunits from recordings of retinal ganglion cells. These include direct fits of cascaded LN models [11,13,46], statistical inference techniques [12,19], tomographic reconstruction [14], and making use of anatomical data [30] or prior determination of photoreceptor locations [15]. Given that subunits are thought to correspond to presynaptic bipolar cells [17–19], these efforts of subunit inference also aim at providing insight into the layout of bipolar cell receptive fields and bipolar-to-ganglion cell connectivity. Yet, how much the precise knowledge of subunit number and position matters for response predictions of computational models of ganglion cells has been debated [20]. Moreover, obtaining sufficient recording data for the challenging task of subunit identification may often be impractical.

For our comparison with the SC model, we implemented a subunit model based on spike-triggered covariance (STC) analysis combined with a probabilistic framework for transforming the resulting filters into spatially localized subunits [16]. We chose this approach for its conceptual simplicity and its straightforward applicability to our spatiotemporal white-noise data, which, despite its relatively coarse spatial resolution, provided sufficient signal for reliable STC computation. It should be noted, however, that different subunit inference methods are likely to yield varying results, and the performance of subunit models relative to the SC model may depend on the specific inference approach employed. For instance, methods that directly optimize subunit filters for response prediction [11–13] or that incorporate anatomical constraints [15,30] might achieve better predictive performance than our STC-based approach. Nonetheless, the comparison of SC and subunit models showed that the SC model is not easily outperformed by taking more detail of nonlinear spatial integration within the receptive field into account.

Besides subunit models, other multi-filter approaches could in principle address nonlinear spatial integration. Standard techniques like spike-triggered covariance analysis [31–33] or most-informative dimensions [40], however, do not easily provide functional, interpretable models on their own [6,16]. An alternative is a likelihood-based model fit, such as in the Nonlinear Input Model [55], which has had success in capturing excitatory and suppressive components as well as ON–OFF interactions of retinal ganglion cells [47], but which has not yet been applied to nonlinear spatial integration within the receptive field center.

Conceptually, the closest alternative to the SC model would be to fit a generalized quadratic model (GQM; [56–58]) to the data. In the GQM, a neuron's activation is described by a quadratic form (general second-order polynomial function) of the stimulus components, such as here the pixel-wise intensity values over space and time. By a nonlinear transformation (typically an exponential function) and a stochastic process (typically Poisson), the activation is then turned into an output spike train, and feedback from generated spikes can be integrated as well via a filtering process that feeds into the model activation. It is the use of a quadratic form of the stimulus components, rather than a linear function as is the case for the generalized linear model (GLM; [37–39]) or for the LN model, which renders the model sensitive to spatial structure beyond the linear activation of the receptive field and thus incorporates effects of receptive-field nonlinearities. This is equivalent to our use of the local spatial contrast as a way to capture nonlinear spatial integration. In fact, the GQM can be

viewed as a more general form of the SC model. This generality, on the other hand, makes fitting the GQM much harder as compared to the more constrained SC model, owing to the many free parameters in the quadratic form, in particular in the spatiotemporal scenario considered here. Thus, this would likely require large amounts of experimental data obtained in long recordings of spiking activity under visual stimulation as well as the exploration of appropriately strong parameter regularization.

## Spatial scale of nonlinear integration

Using the SC model with a spatial stimulus smoothing prior to computing the local spatial contrast allowed us, at least for the case of white-noise stimulation, to determine the spatial scale for assessing receptive-field nonlinearities in the model. Note that this spatial scale is not a single scale at which the nonlinearities occur, but should be understood as a threshold so that fine spatial contrast below this scale is averaged out by linear integration whereas contrast at larger scales can contribute to activation. This is consistent with the concept of subunit models, which average out spatial contrast within individual subunits but are sensitive to contrast that spreads across multiple subunits, thus effectively low-pass filtering spatial contrast with a scale given by the size of the subunits.

In our analysis, the use of the marmoset retina allowed us to identify several known types of ganglion cells and perform this analysis in a cell-type-specific fashion. We found that all four analyzed cell classes yielded a similar scale of around 40–50 μm. This matches typical scales of receptive fields and dendritic trees of presynaptic bipolar cells in the peripheral monkey retina. In macaque, for example, receptive-field sizes of midget bipolar cells seem to lie around 30–50 μm [59], and diffuse bipolar cells, the putative main inputs of parasol ganglion cells, have been reported with dendritic tree sizes, which may be expected to match their receptive fields, around 30–50 μm [60] and receptive fields of 70–110 μm [59]. Note, though, that these measures might differ between macaque and marmoset. For the marmoset retina, data are sparser, but anatomical measurements in the retinal periphery suggest bipolar cell dendritic tree sizes in the range of 20–80 μm [61].

Despite partly comparing across different species (macaque versus marmoset) and across likely different retinal eccentricities as well as different measures (anatomical versus electrophysiological), these numbers indicate that the 40–50 μm spatial scale of nonlinear stimulus integration lies in or near the expected range, given that the spatial contrast sensitivity is thought to result from nonlinear signal transmission between bipolar and ganglion cells. The spatial scales of nonlinear stimulus integration therefore likely reflect a biological constraint rather than a particular evolutionary adaptation to the stimulus statistics of natural scenes, in particular given that the statistics of natural images are largely scale-invariant [34]. Note also that the actual scale in terms of degrees of visual angle in the outside world depends also on the size of the eye, giving, for example, macaque retinal ganglion cells likely a considerably finer sensitivity to spatial contrast as compared to marmoset because of the larger macaque eye [62].

## Nonlinearity of ON and OFF parasol cells

We found that ON parasol cells in our marmoset data displayed stronger spatial nonlinearities than OFF parasol cells (Fig 1D,G and Fig 4B), which matches our previous observations in a different study of the marmoset retina [11]. This contrasts findings in the macaque retina, for which rather linear spatial integration by ON parasol cells has been reported when stimulated with natural or complex scenes [63–65]. Yet, the degree of spatial nonlinearity may not be a fixed cell-type feature, but rather be scene-dependent [20], letting linear spatial integration also fail for macaque ON parasol cells in many examples [7], and depend on stimulus context, such as ambient light intensity [66], surround activation [52], or adaptation state [63].

It also seems plausible that the relative strength of spatial nonlinearities between ON and OFF parasol cells is species-dependent and differs between macaque and marmoset, given their evolutionary divergence approximately 43 million years ago [67]. Among the reported disparities are not only the well-known dichromacy in all male and some female

marmosets [68] but also, for example, the density of peripheral cone photoreceptors and their convergence onto parasol cells [27] as well as the expression of certain transcription factors in non-parasol and non-midget ganglion cells [69,70]. Furthermore, the ON and OFF pathways distinctly differ for macaque versus marmoset in stainings for choline acetyltransferase (ChAT), with macaque retinas displaying considerably more cells marked in the ON ChAT band than the OFF ChAT band and vice versa in marmosets [71].

### Artificial versus naturalistic stimuli

It may seem surprising that the analysis under natural stimuli did not yield an optimal scale of nonlinear spatial integration. Rather, stimulus smoothing before computing the local spatial contrast generally had no or a detrimental effect. The likely reason is that natural scenes, such as those in the applied naturalistic movie, are dominated by low spatial frequencies, owing to the approximate $1/f^2$-scaling of the power spectrum of light intensities in natural images with spatial frequency $f$ [34,35]. Thus, smoothing at small spatial scales, e.g., below the scale of receptive-field subunits, does little to the stimulus and leads to no appreciable change in the model's prediction accuracy, unlike in the case of white-noise stimulation. This does not mean, of course, that local spatial contrast and nonlinear stimulus integration are not important in the case of natural stimuli — the comparison with the LN model demonstrates that they are, at least for ON and OFF parasol and Large OFF cells — but the relevant spatial structure for natural stimuli is found at larger spatial scales, mostly just below the scale of receptive fields themselves where stimulus power is sufficiently large and smoothing at smaller scales does not affect this spatial structure.

These results suggest that, despite the many advantages of using naturalistic stimuli to analyze neuronal coding, specific aspects of stimulus processing might still be easier to investigate with artificial stimuli that emphasize the stimulus feature of interest, such as spatial contrast at high spatial frequency.

The relative lack of high-spatial-frequency signals in the naturalistic movie as compared to the white-noise stimulus is also apparent in the reduced importance of local spatial contrast in the SC model under the naturalistic movie. Performance gains relative to the LN model and values of the parameter for weighting the local spatial contrast contribution were generally smaller than under white noise. This may simply reflect that responses are, on average, determined more by the light intensity signals inside the receptive field than by spatial contrast, so that the model naturally profits from putting more weight on the former when determining its activation.

On the other hand, it also seems feasible that the retinal neurons adapt to the different stimulus statistics under naturalistic versus white-noise stimulation [48,49] and that changes in the models between those two scenarios reflect actual changes in nonlinear spatial integration, as can be observed for other alterations of stimulus context [52,63,66]. Yet, our analysis of how firing rates depend jointly on $I_{mean}$ and LSC (Fig 1) suggests that nonlinearities of spatial integration under natural stimuli are comparable to those under white-noise stimulation. Future investigations may aim at investigating this in more detail by adding appropriate probe stimuli to the artificial and naturalistic stimulation or by frequently switching between the two, as done to analyze other complex adaptation phenomena [72]. Finally, applying different types of naturalistic stimuli, for example, containing different levels and types of motion signals, and comparing model performance and the importance of spatial-contrast information under these scenarios may help understand how general or specific the observed effects are.

## Methods

### Ethics statement

All experimental procedures were performed in strict accordance with national and institutional guidelines. Electrophysiological recordings from marmoset retinas were performed on tissue obtained from animals used by other researchers, as approved by the institutional animal care committee of the German Primate Center and by the responsible regional government office (Niedersächsisches Landesamt für Verbraucherschutz und Lebensmittelsicherheit, permit number 33.19-42502-04-20/3458).

## Electrophysiological recordings

We used two retinas from male adult common marmosets (*Callithrix jacchus*). The eyes were enucleated and the cornea, lens, and vitreous humor were carefully removed immediately after the sacrifice of the animal. Eyecups were transported to the electrophysiology lab in a light-sealed container filled with freshly oxygenated (95% $O_2$ and 5% $CO_2$) Ames' medium (Sigma-Aldrich), supplemented with 6 mM D-glucose and buffered with 20 mM $NaHCO_3$ to maintain a pH of 7.4. Retinas were then dark-adapted for 1–2 hours in a light-sealed container while submerged in continually replenished oxygenated Ames' medium. Care was taken to avoid oxygen depletion and to reduce the amount of light incident onto the retinal tissue to a minimum.

Each retina was first hemisected along the superior-inferior axis. For recording, a smaller piece, a few millimeters wide, was cut off, carefully separated from the sclera, choroid, and pigment epithelium, and gently flattened onto a transparent dialysis membrane held taut across a hollow, plastic support, with the photoreceptor side of the tissue towards the membrane. The support, along with the membrane and retinal piece, was slid into a plastic bath chamber, glued onto a planar microelectrode array (MEA; MultiChannel Systems) and designed to snugly fit the support. Thereby, the inverted support positioned the retinal tissue on top of the recording electrodes with ganglion cell-side down. MEAs had 252 electrodes with 10 μm electrode diameter and 60 μm electrode spacing and were coated prior to experiments with a thin layer of poly-D-lysine, which helped keep the retinal tissue in place. The support with the membrane was carefully removed, leaving the retina piece on the MEA, and replaced with a support without a membrane, but with a small horizontal glass plate a few millimeters above the retinal tissue. This minimized refractive effects of the stimulus light when the bath chamber was filled with perfusion solution up to the lower surface of the glass plate. The MEA was then transferred to the recording setup and connected with an amplifier and analog-to-digital converter (MultiChannel Systems). Once mounted, the retina was perfused with the oxygenated Ames' medium at a rate of around 400–450 ml/h while being held at a temperature of 32–34° C with the help of an inline heater and a heating element under the array. The remaining retina pieces were stored for later experiments in a light-sealed container at the same temperature range under constant perfusion with the Ames' medium.

The recorded extracellular signals were amplified, bandpass-filtered between 300 Hz and 5 kHz, and stored digitally at a sampling rate of 25 kHz. Spikes were extracted using a modified version of the Kilosort spike-sorting algorithm [73,74], modified code available at https://github.com/dimokaramanlis/KiloSortMEA). The resulting clusters were manually curated using the software package Phy2 (https://github.com/cortex-lab/phy). Only well-isolated clusters with minimal refractory period violations were chosen for further analysis, and each such cluster was deemed to correspond to a single cell. For the purpose of modeling, spike times of each cell were binned at the resolution of the projector refresh rate (85 Hz).

## Visual stimulation and basic cell characterizations

Visual stimuli were rendered by a custom-made software, written in C++ and OpenGL. The stimuli were displayed on a gamma-corrected monochrome grayscale OLED monitor (eMagin) at a refresh rate of 85 Hz and a resolution of 800x600 pixels. A telecentric lens (Edmund Optics) was used to project the stimuli onto the retina, resulting in a pixel size of 7.5 μm x 7.5 μm on the retinal surface. The mean light level on the surface of the retina was either 5.5 mW/m$^2$ or 4.6 mW/m$^2$, depending on the specific recording setups. Based on photoreceptor spectral sensitivity profiles [75], peak sensitivities of marmoset rods and cones [76,77], and collecting areas of 0.37 μm$^2$ for cones and 1.0 μm$^2$ for rods, here applying values from macaques [78,79], we estimated the isomerization rates per photoreceptor and second for the applied range of mean light intensities as 380–450 for S-cones, 2,400–2,900 for M-cones, and 7,200–8,600 for rods.

**Spatiotemporal white-noise stimulus.** We stimulated each retina with a spatiotemporal white-noise stimulus, which consisted of a sequence of frames of black and white squares (100% Michelson contrast) arranged in a uniform checkerboard pattern of 150x200 stimulus squares, with each square spanning 30 μm x 30 μm on the surface of the retina. The contrast values for each square were independently and randomly obtained from a binary distribution and

updated at the monitor refresh rate of 85 Hz. For analysis, the stimulus was denoted by the Weber contrast values +1 or −1 of each pixel. The stimulus was divided into trials, each consisting of a 150 s- or 300 s-long (value depending on the particular recording) distinct, non-repeating part, which was used for model training, and a 30 s- or 60 s-long fixed part ('frozen noise'), which was repeated in each trial and used for model testing. Total recording duration with this stimulus in each recording was either about 30 min or 60 min.

**Estimation of receptive fields and stimulus filters.** We estimated the receptive field of each cell using the responses to the training set of the spatiotemporal white-noise stimulus. We computed the spatiotemporal spike-triggered average (STA) from the binned responses (85 Hz) over the past 30 stimulus frames (corresponding to 353 ms). Spikes during the first 29 frames in each trial were excluded in this computation.

From the spatiotemporal STA, we extracted a spatial and a temporal component in the following way, as previously described [11]. For each STA, we selected pixels whose absolute peak intensity over time exceeded six times the robust standard deviation of the absolute peak intensities of all pixels around their median. (The robust standard deviation is defined as 1.4826 times the median absolute deviation and is a consistent estimator of the standard deviation of a normal distribution.) The temporal component was taken as the average of the time course of the selected pixels in the STA. The spatial component was computed by projecting the three-dimensional STA onto the temporal component. The receptive field center was estimated by determining the pixel in the STA with the largest variance along the temporal dimension. Lastly, the spatial component of the STA was cropped to a region of 40x40 stimulus squares (1200 μm x 1200 μm) around the receptive field center. The spatial and the temporal components were normalized by their respective Euclidean norms.

For modeling responses to the naturalistic movie stimulus, we followed the same procedure as above to obtain the STA using spatiotemporal white noise and then upsampled the spatial dimensions of the STA from 200x150 squares of the checkerboard layout to 800x600 pixels, the resolution of the movie stimulus, before extracting the spatial and temporal components. The upsampling was performed by simply using the same STA value for all pixels in the region of a given white-noise stimulus square. The resulting spatial component was again cropped around the receptive field center to a region of 160x160 pixels (1200 μm x 1200 μm). We then fitted a two-dimensional elliptical Gaussian function to each spatial component and used the fit as a spatial filter. The fitting was done using the `curve_fit` function of Python's `scipy` library. Pixels beyond a 3-$\sigma$ boundary around the center of the Gaussian fit were set to zero. From the Gaussian fit, we also estimated the receptive-field size as the diameter of a circle that has the same area as the 1.5-$\sigma$ ellipse of the fit.

**Autocorrelograms.** To aid cell-type classification, we computed the autocorrelogram of the spike times for each cell, based on its responses to the training segments of spatiotemporal white noise. We used a bin size of 0.5 ms, denoting the occurrence of a spike by a value of unity and no-spike by zero, and retained 150 bins, corresponding to a range of temporal lags of 0–75 ms. Each autocorrelogram was normalized to unit sum.

**Naturalistic movie stimulus.** To create a naturalistic movie stimulus, we utilized a short, publicly accessible science-fiction film released by the Blender Foundation under the CC BY 3.0 license [28] and made available for download as individual movie frames in the Portable Network Graphics (PNG) format at https://media.xiph.org/tearsofsteel/tearsofsteel-1080-png/. The movie was chosen for its diverse array of textures and light intensities. For experiments, the movie frames were sequentially projected onto the retina. Excluding frames from the movie's introductory and credit sequences, as well as extended periods of homogeneous black, left us with 14,544 movie frames, equivalent to approximately 10 minutes of the movie, given the original frame rate of 24 Hz. Of these, 1,434 continuous frames were reserved as a test stimulus, with the remaining 13,110 movie frames constituting the training set. The test segment was chosen through visual inspection, focusing on scenes featuring a balanced mix of fast and slow moving objects and diverse textures. To reduce memory requirements, we cropped each movie frame from a resolution of 1920x800 pixels to 1000x800 pixels, by removing 460 pixels from each end along the horizontal axis.

The movie's frame rate of 24 Hz differed from the projector's refresh rate of 85 Hz. For presentation to the retina, we therefore upsampled the cropped frames by a factor of 3.54 by repeating individual frames three to four times, randomly

selecting for each frame the presentation duration of three or four projector frames with probabilities of 0.46 and 0.54, respectively. Each cropped frame was converted from the RGB color space to grayscale using a weighted sum of R, G, and B values to compute linear luminance as $0.2126\,R + 0.7152\,G + 0.0722\,B$.

To generate a unique, long stimulus sequence from a short movie and to increase the diversity of stimulus patterns within the receptive fields of individual ganglion cells, we manipulated the stimulus in the following ways. First, we repeated the grayscale frames of the training set as many times as needed to yield a training set size of 255,000 stimulus frames, separated into 10 trials of 300 s each. Next, we added gaze-shift-like global motion signals to the stimulus by shifting the center of each frame independently using simulated eye movements (see below). Occasionally, due to the overlayed movement, the shifted frame exceeded the bounds of the cropped movie frame, leading to the replacement of missing pixels with gray pixels at the edges of the presentation region, but on the retina this did not overlap with the region from which ganglion cells were recorded. Lastly, to add further variability, 40% of the training segments were flipped vertically. The test stimulus sequence was treated in the same way as the training stimulus and repeated 10 times (interleaved with the training stimulus sequences), always with the same trajectory of simulated eye movements and sequence of frame repeats and not flipped.

The final stimulus was projected onto the retina at a resolution of 800x600 pixels, with each pixel covering 7.5 µm x 7.5 µm on the surface of the retina, and a mean light level of 76% of the white-noise stimulus in order to accommodate for the range of pixel intensities and the limited maximal intensity of the projector. The root-mean-square contrast of the movie over space and time was 45% of the mean intensity.

For analysis, we converted the light intensity values of each pixel in the stimulus to Weber contrast values by subtracting its mean intensity over time and subsequently normalizing by this mean intensity.

**Simulated eye movements.** We introduced global motion to the movie stimulus by spatially shifting the center position of each frame, mimicking saccades, fixations, and fixational eye movements, consistent with the statistics of active vision in marmosets [24]. Saccades were characterized by a combination of amplitude, direction, and duration. Saccade amplitudes were randomly sampled from an exponential distribution with a scale parameter of 200 µm. The directions of saccades were randomly drawn from a uniform distribution between 0 and 360 degrees. Given that the displayed movie's frame rate was constrained by the monitor's refresh rate, saccade duration was sampled from three possible values (23.5 ms, 35.3 ms, 47.1 ms), corresponding to 2, 3, and 4 frame updates, with probabilities of 0.35, 0.4, and 0.25, respectively. During these periods the frame's center position was linearly shifted from the original position to the target of the saccade.

Fixation durations were randomly sampled from an exponential distribution with a refractory period. This type of distribution was originally proposed as a model for fixation times in humans [80] and has since been used by others to describe the statistics of fixation duration in other animal models [81,82]. We here applied a scale parameter of 200 ms and a refractory period of 100 ms as used previously for primate-retina recordings [82]. Additionally, to simulate fixational eye movements, the center of each cropped frame was jittered in the $x$ and $y$ directions for every frame update during periods of fixation at 85 Hz. The amplitude of the jitter was sampled independently for each direction from a normal distribution with a mean of zero pixels and a standard deviation of two pixels.

Simulated eye movement traces were generated in chunks of 10 s for both the training and the test segments. The fixation center was reset to the center of the monitor at the beginning of every chunk, preventing the fixation center from drifting out of the cropped frame. Additionally, chunks where the fixation center drifted more than 200 monitor pixels in any direction were rejected and regenerated.

**Reversing-grating stimulus and index.** We also recorded responses of RGCs to a standard reversing grating, a stimulus widely used to probe spatial nonlinearities of the cells. Here, the stimulus consisted of alternating vertical black and white stripes of a fixed width (100% Michelson contrast; mean intensity same as for white-noise stimulus) that change polarity every 500 ms. We used gratings with stripe widths of 1, 2, 4, 8, 16, 32, 64, and 800 pixels and a range of spatial phase shifts, which depended on the applied stripe width. Each grating was shown for 12 temporal periods of the reversal, and the whole sequence was repeated twice. Responses were binned in 5 ms windows for further analysis.

To compute the reversing-grating index for each cell, we averaged its binned responses across all 48 repeats for each combination of stripe width and phase shift to yield 1 s-long peri-stimulus time histograms (PSTHs). We computed the amplitude of the harmonics of each PSTH using a Fourier transform and extracted the amplitude $F_1$ at the stimulus frequency (1 Hz) and the amplitude $F_2$ at twice the stimulus frequency (2 Hz). We extracted the largest $F_2$ and the largest $F_1$ across all widths and phases for each cell, aiming at capturing the maximum mean-light-intensity-induced and maximum spatial-contrast-induced activity modulations, and took the ratio $F_2/F_1$ to be the reversing-grating index for that cell [83].

## Cell selection

Since model fits will be compromised if responses are unreliable across trials, we selected cells based on the reliability of their responses to the test stimuli. We evaluated reliability separately for the white-noise stimulus and the naturalistic movie and required cells to pass reliability criteria for both stimuli.

We used two measures of reliability. The first was the fraction of explainable variance $\text{FEV} = (\sigma^2_{total} - \sigma^2_{noise})/\sigma^2_{total}$ [84], used as a measure of the variability in the data that cannot be attributed to noise and is thus considered a reliable signal. Here, the total variance $\sigma^2_{total}$ is the variance of spike counts per time bin across all time bins and trials, and the noise variance $\sigma^2_{noise}$ is the spike-count variance computed across trials for a given time bin, averaged across time bins.

The second measure was the coefficient of determination ($R^2$) between the binned firing-rate responses to odd versus even trials of the test set. Here, we used a symmetrized version of the coefficient of determination [83], where the average responses to either the odd or the even trials were taken as the prediction of the other and the average of both versions was used.

Based on visual inspection, we selected conservative thresholds and discarded cells with FEV less than 0.15 or with negative $R^2$. In total, 180 recorded cells passed our reliability criterion and were included for further analyses.

## Cell-type classification

We used a semi-automated approach to identify cell types in our dataset. To start, we constructed a feature vector for each reliable cell, which consisted of the 30 elements of the temporal filter, the receptive-field size, and the scores corresponding to the first ten components from a principal component analysis (PCA) of the autocorrelograms (with each entry Z-scored across cells) of all cells. After Z-scoring each entry in the feature vector independently across cells, we reduced the dimensionality of the feature vectors using PCA, retaining as many components as necessary to explain 90% of the variance (26 components). Finally, we used KMeans++ on the reduced-dimensionality feature vectors to determine 6–8 initial clusters of cells. These clusters were manually curated to ensure each cluster had similarly-sized receptive fields that tiled the retinal surface and showed consistency in the shapes of the temporal filters and autocorrelograms. We used functions from the Python library `scikit-learn` [85] to run the PCA and KMeans++ algorithms.

We identified three clusters as corresponding to OFF midget cells (n = 51), OFF parasol cells (n = 69) and ON parasol cells (n = 35), according to receptive-field size and temporal filters. For analysis, we selected three additional clusters with fairly consistent features and containing multiple cells and grouped these into a single class of Large OFF cells (n = 15). This multi-cluster class is not considered to correspond to a single cell type. Cells not included in one of these groups were discarded as they either displayed unreliable receptive field measurements or came from clusters with only few samples.

## Two-dimensional firing rate histograms and LSC sensitivity

We investigated the combined influence of the mean light intensity and the high-frequency spatial contrast in a cell's receptive field on the cell's firing response using two-dimensional firing rate histograms. For each cell and each stimulus condition (white noise and naturalistic movie), we first computed two corresponding measures, the weighted mean intensity ($I_{mean}$) and the local spatial contrast (LSC), for each frame of the stimulus (described below). Then, we binned both

signals independently and constructed two-dimensional firing rate histograms by averaging the cell's firing response over all time points that corresponded to each pair of $I_{mean}$ and LSC bins. For visualization, we used 40 equally-spaced bins for each signal and normalized the entries to the largest entry in each histogram.

To quantify how the LSC affected the firing rate in these histograms, we aimed at measuring by how much the firing rate-versus-$I_{mean}$ curves in the rows of the histogram were shifted relative to each other, depending on the LSC value. For this purpose and to have a better representation of the histogram values near the edges, we recomputed the histograms by binning both $I_{mean}$ and LSC independently using 60 bins, such that for each of the two signals, all the corresponding bins had about the same number of values. We excluded the first and last bins for each signal in further computation and used the remaining bins to construct two-dimensional firing rate histograms as described above. In effect, the rows of the histogram describe changes in the firing response of the cell with increasing mean light intensity at different levels of spatial contrast in the receptive field. We used the softplus function parametrized as

$$f_i(x) = a \cdot \log_e \left[ 1 + \exp \left( b \cdot (x - c_i) \right) \right], i = 1, ..., N \tag{1}$$

to fit a family of curves to the $N$ rows of the histogram, with parameters $a$ and $b$ common across the rows and the parameters $c_i$ depending on the rows $i$. These can be viewed as LSC-dependent threshold values of mean luminosity beyond which the cell's firing response rapidly increased.

We observed that the obtained $c_i$ values depended approximately linearly on the LSC level, $l_i$, and used the linear relationship between the values of $l_i$ and of $c_i$ as a measure of the cell's sensitivity to LSC. To ensure a robust estimate of the linear relationship, we removed outlier pairs $(c_i, l_i)$ using the Local Outlier Factor score with 5 nearest neighbours (as implemented in the Python library `scikit-learn` [85]). Then, we applied linear regression to determine the gradient $m$ of the straight line fit to all remaining points, taking the $c_i$ values as independent variables and the $l_i$ values as dependent variab/es. Finally, we defined our metric as LSC sensitivity = $-1/m$ in order to obtain a measure where larger, positive index values correspond to more nonlinear cells.

## Modeling

We developed a spatiotemporal version of the spatial contrast (SC) model [21] to predict responses of individual ganglion cells to spatiotemporal white noise and natural stimuli. The model takes spatial contrast inside a cell's receptive field into account by providing a measure of the pixel-wise intensity range as an input component and thus aims at capturing non-linear spatial integration. For comparison, we employed a classical linear-nonlinear (LN) model, which is based on linear spatiotemporal filtering and thus not sensitive to high-frequency spatial contrast inside the receptive field.

**Linear-nonlinear (LN) model.** The LN model involved a linear filtering operation of the stimulus by the cell's temporal and spatial filters and a subsequent nonlinear activation function, describing the relationship between the linear-filter output and the cell's firing response. We separated the spatiotemporal filter of a cell into a temporal and a spatial filter and obtained each, as described above, from the STA of the cell under spatiotemporal white noise. For filtering, we first convolved the stimulus $z_{st}$ (denoting the Weber contrast of a pixel with spatial index $s$ at time bin $t$) pixel-wise with the cell's temporal filter $v_t$ of length $M$ to obtain the convolved stimulus $h_{st}$:

$$h_{st} = \sum_{\tau=0}^{M-1} z_{s(t-\tau)} \cdot v_\tau. \tag{2}$$

We then applied the spatial filter $u_s$, taken as the Gaussian fit to the spatial component of the receptive field, by computing the scalar product of the pixel-wise elements of the stimulus and the filter to obtain the linear activation $x_t$ of the LN model at time bin $t$:

$$x_t = \sum_s u_s \cdot h_{st}.$$

(3)

Finally, the filtered signal was transformed by the nonlinear function $f(\cdot)$ to obtain the model's expected spike-count response $\lambda_t^{LN}$:

$$\lambda_t^{LN} = f(x_t).$$

(4)

We here took the output nonlinearity $f$ to be a softplus function, parameterized as

$$f(x) = a \cdot \log_e \left[ 1 + \exp(b \cdot x + c) \right]$$

(5)

The parameters $a$, $b$, and $c$ were learned during model fitting. The details of the fitting procedure are described below.

**Spatial contrast (SC) model.** In our version of the SC model, the temporal component of the stimulus is accounted for by convolving it with the cell's temporal filter as in Eq 2. From the convolved stimulus, we computed for every time point $t$ the mean light intensity signal $I_{\text{mean } t}$ and the local spatial contrast $LSC_t$ within the receptive field of the cell and combined them linearly. $I_{\text{mean } t}$ is the linearly filtered stimulus, similar to Eq 3, defined as a weighted average of the (temporally convolved) stimulus $h_{st}$, with weights coming from the spatial filter $u$:

$$I_{\text{mean } t} = \frac{\sum_s u_s \cdot h_{st}}{\sum_s u_s}.$$

(6)

$LSC_t$ is analogously obtained as a weighted standard deviation of the pixel intensities from the (temporally convolved) stimulus $h_{st}$:

$$LSC_t = \sqrt{\frac{\sum_s u_s \cdot (h_{st} - I_{\text{mean } t})^2}{\sum_s u_s}}.$$

(7)

Note that this definition of LSC differs from the original one suggested in [21], where LSC was simply defined as a standard deviation of the spatially filtered stimulus. For large receptive fields relative to pixel size, as was the case in [21], the two definitions are nearly equivalent. However, for smaller receptive field, as we encounter for the cell types of the primate retina analyzed here, Eq 7 is superior, as it is not sensitive to variance of pixel intensities induced by the spatial filtering.

To obtain the model response from $I_{\text{mean}}$ and LSC, a free parameter $w$ was used to determine the relative importance of spatial contrast information for each cell. Finally, an activation function, again taken as a parameterized softplus function, modeled the relationship between the linearly combined signal and the expected spike-count response of the cell, similar to the LN model:

$$\lambda_t^{SC} = f(I_{\text{mean } t} + w \cdot LSC_t).$$

(8)

Together, the SC model has four free parameters – $a$, $b$, and $c$ for the shape of the output nonlinearity (Eq 5) and the weight factor $w$. Optimization of the parameters is described below.

Note that, equivalently, one could also use weights for both $I_{\text{mean}}$ and LSC with an appropriate constraint, e.g., $w_1$ and $w_2$ with $w_1 + w_2 = 1$, which may help avoid very large $w$ values for cells whose responses would be mostly determined by LSC

with little influence by $I_{mean}$. Here, we neither expect nor observe such cells and therefore use the weighting introduced above for simplicity and for staying close to an LN model with added LSC sensitivity.

**Subunit model.** The subunit model we implemented for comparison with the SC model has a standard linear-nonlinear-linear-nonlinear (LNLN) cascade structure. The first LN stage consists of multiple spatial filters representing the subunits, each normalized to unit Euclidean norm. Similar to the LN and the SC models above, to apply the model obtained for a specific cell to a stimulus, we first convolved the stimulus with the cell's temporal filter extracted from the spatiotemporal STA and then projected the temporally convolved stimulus onto each subunit filter. Next, we half-wave rectified the activation signals from the individual subunits and passed these to the second LN stage, where we computed their weighted sum and transformed this sum using an output nonlinearity.

Concretely, the model activation at every time bin $t$ was thus given as,

$$\lambda_t^{sub} = f^{sub}\left(\sum_{i=1}^n p^i \cdot \text{ReLU}\left(\sum_s q_s^i \cdot h_{st}\right)\right),$$

(9)

where $i$ enumerates the subunits, $h_{st}$ is the temporally convolved stimulus, $p^i$ and $q_s^i$, respectively, are the weight and the spatial filter of subunit $i$, ReLU is the subunit nonlinearity specified as a Rectified Linear Unit function, and $f^{sub}$ is the output nonlinearity, here a softplus function parametrized as

$$f^{sub}(x) = a \cdot \log_e\left[1 + \exp(x + c)\right].$$

(10)

The subunit activation weights, $p$, and the parameters $a$ and $c$ of the output nonlinearity are the free parameters of the subunit model. Note that the softplus function here contains no scaling parameter $b$ to be multiplied with $x$ because the subunit activation weights already provide for appropriate scaling of the input $x$.

Following [16], we determined the spatial filters of the subunit model $q_s$ for each cell by first extracting an orthogonal basis of the relevant stimulus space from the eigenvectors of the spike-triggered covariance (STC) matrix [31–33]. To reduce stimulus dimensionality for the STC analysis, we spatially cropped the stimulus for each cell to the smallest square region that enclosed the 3-$\sigma$ contour of a 2-dimensional Gaussian fit to the cell's spatial filter, up to a maximum side length of 16 stimulus pixels of the spatiotemporal white-noise stimulus. Since the eigenvalue spectrum of the STC analysis typically did not indicate a clear cutoff for the relevant stimulus dimensions, we always used the top 16 eigenvectors from the STC analysis (measured by the absolute deviations of the corresponding eigenvalues from the median eigenvalue) for further analysis. We found that this provided a sufficiently rich subspace to represent the relevant stimulus features, as increasing the number of used eigenvectors did not improve the subunits or the subunit model performance.

The set of STC eigenvectors was then linearly transformed by optimizing a probabilistic spiking model, which we call the logical-OR model, as it approximates the combination of filters by a logical-OR operation. This type of transformation of the STC eigenvectors has been shown to yield subunit-like filters [16]. The logical-OR model aimed at optimizing the probability $P_t^{OR}$ of a spike occurring within a time bin $t$ given the temporally convolved stimulus $h_{st}$ using the following approximated logical-OR operation:

$$P_t^{OR} = 1 - \prod_{i=1}^n \left[1 - g^i(U^\top A^i \cdot h_{st})\right],$$

(11)

where $i$ indexed the $n$ subunits, the rows of $U$ contained the STC eigenvectors, and $A^i$ was the weight vector for subunit $i$. The nonlinearity $g^i$ in the model was the logistic function, parametrized as

$$g^j(x) = \frac{1}{1 + e^{-(\alpha^i + x)}},$$

(12)

where $\alpha^i$ sets the activation threshold for subunit $i$. The weight matrix $A$ and the activation thresholds $\alpha$ were the tunable parameters of the logical-OR model. Each resulting functional subunit $q^i = U^\top A^i$ was normalized to unit Euclidean norm and used as the spatial filter for subunit $i$ in the subunit model described earlier (Eq 9). The number of functional subunits $n$ need not match the number of used STC eigenvectors; instead it was used as a hyperparameter and optimized by a process of pruning at two stages. This pruning, the binning of the spike trains, as well as the fitting of other model parameters are described below.

**Model fitting and evaluation.** For all models described above, the responses of the cells were binned by the update times of the stimulus (85 Hz). Model parameters of the LN model, the SC model, and the subunit model were optimized by a maximum-likelihood fit under the assumption that spike counts per bin are determined according to a Poisson process with the expectation value given by the model response for that bin, $\lambda_t^{LN}$ (Eq 4), $\lambda_t^{SC}$ (Eq 8), and $\lambda_t^{sub}$ (Eq 9) for the three models, respectively. The probability under each of these models of observing the complete binned spike train with spike counts $n_t$ is given by the product of the probabilities for individual bins:

$$P(\{n_1, n_2, \ldots\}) = \prod_t \frac{\lambda_t^{n_t} e^{-\lambda_t}}{n_t!}$$

(13)

where $\lambda_t$ is either $\lambda_t^{LN}$, $\lambda_t^{SC}$, or $\lambda_t^{sub}$. This yields the negative log-likelihood function (up to a constant)

$$\mathcal{L}(\Theta) = -\sum_t n_t \ln \lambda_t + \sum_t \lambda_t,$$

(14)

where $\Theta$ represents the parameters of the model to be optimized. In the case of the LN model, these were the $a$, $b$, and $c$ from the softplus nonlinearity and in the case of the SC model, additionally the weight $w$. The subunit model parameters comprised the softplus nonlinearity parameters $a$ and $c$ as well as the subunit weights $p$.

For the logical-OR model, which provides a spike probability as output, we binarized the spike trains before training by replacing the spike count in all non-zero spike bins with unity. We optimized the free parameters of the model – the activation thresholds $\alpha$ and the weight matrix $A$ – by minimizing the negative log-likelihood of the Bernoulli distribution, which is given by

$$\mathcal{L}^{OR}(\Theta) = -\sum_t \left[ n_t \ln P_t^{OR} + (1 - n_t) \ln(1 - P_t^{OR}) \right],$$

(15)

where $\Theta$ represents the parameters of the logical-OR model to be optimized and $n_t \in [0, 1]$ are the binarized responses.

For the LN and SC models, we minimized Eq 14 with the `minimize` function provided in Python's `scipy` library [86], applying the `L-BFGS-B` algorithm, a quasi-Newtonian constrained optimization algorithm that supports upper and lower bounds on variables, which we found to provide reliable solutions for our problem. Before fitting each of these two models, we Z-scored the activation signal obtained from the spatiotemporal filtering in the LN model as well as the I$_{mean}$ and the LSC signals for the SC model by their respective means and standard deviations. This allowed us to apply a simple, largely common set of initial parameters for optimization of each model across all cells. We used the following initial values for the model parameters: $a = \max(n_t)$, $b = 1.0$, $c = -2.0$, as well as $w = 0.0$ in the case of the SC model. Since the firing response of a cell cannot be negative, a lower bound of zero was set for the parameter $a$ in both models by passing it as an argument to the `minimize` function.

For the logical-OR model and the subunit model, the negative log-likelihood functions were minimized using routines provided by Python's `PyTorch` library [87]. Both models were trained on spatiotemporal white-noise via mini-batch stochastic gradient descent with the `Adam` optimizer and a batch size of 512 on the non-repeating segments from 80% of the trials. The remaining 20% of the non-repeating segments were used as a validation set to evaluate the model during training. Learning rates were set to $5 \times 10^{-3}$ for the logical-OR model and $1 \times 10^{-3}$ for the subunit model and were dynamically reduced by a factor of 0.8 whenever the validation loss failed to improve for 5 consecutive epochs. The models were trained for a maximum of 500 epochs and training was terminated early if the validation loss did not improve for 50 epochs. The model state corresponding to the lowest validation loss was retained. The subunit weights $p$ were constrained to be non-negative by setting negative values to zero after each gradient update.

The logical-OR model was initialized with 16 functional subunits $q^i = U^\top A^i$, a deliberately generous number chosen as supported by the fact that we generally obtained optimal subunit numbers much smaller than this in the subsequent analysis. The initialization allowed determining the appropriate subunit number automatically via regularization rather than by manual parameter search. To encourage the model to use fewer subunits, a Group Lasso penalty was applied to the functional subunits, penalizing the sum of their $\ell_2$-norms $\sum_i \|q^i\|_2$ with a regularization strength of $10^{-4} \times \sqrt{N_f/400}$, where $N_f$ denotes the number of stimulus pixels in each subunit filter. This scaling ensured that the effective regularization strength remained comparable across cells with different spatial crop sizes. To allow the model to first converge to a reasonable solution before the penalty drives subunit norms toward zero, the regularization term was applied only after a warm-up period of 50 epochs. After training, subunits whose $\ell_2$-norms fell below 10% of the maximum norm were pruned, and the remaining subunits were normalized to unit $\ell_2$-norm before being passed to the subunit model. The subunit weights $p$ were initialized to the pre-normalization norms of the corresponding subunits, preserving the overall contribution of each subunit.

Training of the subunit model proceeded in two stages. In the first stage, an $\ell_1$-penalty was applied to the subunit weights $p$ with a regularization strength of $10^{-4}$. After convergence, subunits whose weights fell below 10% of the maximum weight were pruned. In the second stage, the model was reinitialized with the retained subunits and their corresponding weights, and retrained without regularization until convergence.

For each cell, model predictions from the LN model, the SC model, and the subunit model were evaluated against the average response across trials of the test set using the Pearson correlation coefficient $r$.

## Spatial smoothing and optimal scale of spatial integration

We used a modified version of the SC model to determine the scale at which spatial nonlinearities in the ganglion cell receptive fields occur. This means that spatial contrast on finer spatial scales is averaged out through linear stimulus integration, whereas spatial contrast on broader scales can contribute to activating the cell through the spatial-contrast component of the model. In this modified version of the model, the temporally convolved stimulus ($h_{st}$, Eq 2) was smoothed before extracting the $\text{LSC}_t$ from it. The computation of $\text{I}_{\text{mean } t}$ remained unaffected. The smoothing was performed by convolving $h_{st}$ with a circular Gaussian filter. To probe different scales of spatial smoothing, we applied different standard deviations of the Gaussian filter and defined the scale of a given spatial smoothing as three standard deviations of the filter. This definition of scale was chosen so as to be in line with our definition of receptive-field size, as the applied 1.5-$\sigma$ contour of the receptive field would yield a size of three standard deviations for a circular receptive field. To perform the smoothing, we used the function `gaussian_filter` provided by the Python library `scipy`, with the diameter of the applied region of the filter taken to be six standard deviations of the filter.

For both white noise and naturalistic movie, we tested 20 different values of standard deviation of the smoothing filter, ranging from 0.8 to 12.0 pixels (in steps of 0.4 pixels from 0.8 to 6.4 and then 7.2, 8.0, 8.8, 10.4, and 12.0 pixels), corresponding to spatial scales of 18–270 μm. In each case, model parameters were optimized independently. The change in model performance compared to the case without stimulus smoothing was quantified for each cell as the

ratio of the two corresponding Pearson's $r$ correlation values between prediction and response. The optimal smoothing scale for a cell was determined as the spatial scale where this ratio was maximized. To obtain the optimal scale and the corresponding maximum value at finer resolution, we interpolated between the three datapoints around the largest measured ratio of correlation values using cubic spline interpolation at a resolution of 0.1 μm before identifying the maximum.

## Power spectrum analysis

To illustrate the effect of stimulus smoothing, we examined the radially-averaged power spectral density (RAPSD) of each of the two stimuli after temporal filtering and for various smoothing scales. We analyzed a square region around the center for each stimulus, resulting in a resolution of 600x600 pixels for both the naturalistic movie and the white-noise stimulus. The white noise frames were upscaled from the 150x150 stimulus squares to 600x600 pixels without interpolation. Using a sample OFF parasol cell's temporal filter, we computed the temporally convolved stimulus to account for any effects of temporal filtering on the power spectrum. Stimulus smoothing was here applied as described earlier for the analysis of the spatial scale of nonlinear stimulus integration. For each resulting $N$x$N$ stimulus frame, $h_{mn}$ (where $m$ and $n$ index the stimulus pixels in the two spatial dimensions and the temporal dimension is left out for convenience), we then obtained the discrete two-dimensional Fourier transform $\hat{h}_{kl}$ as:

$$\hat{h}_{kl} = \frac{1}{N} \sum_{m=0}^{N-1} \sum_{n=0}^{N-1} h_{mn} \exp\left\{-2\pi i \frac{mk + nl}{N}\right\}, k = -N/2, ..., N/2; l = -N/2, ..., N/2$$

(16)

where $N$ is the number of pixels in each dimension (here 600 pixels). We then computed the power $|\hat{h}_{kl}|^2$ in each $\hat{h}_{kl}$ component. Next, we binned the two-dimensional power spectrum using radial bins of unit width, centered on the zero-frequency term, and computed the mean power in each bin, yielding the RAPSD for each frame. Finally, we averaged the RAPSDs across all frames for each individual stimulus.

## Acknowledgments

We thank Fred Rieke for advice on experiments with primate retina and Fernando Rozenblit for help with the stimulus generation.

## Author contributions

**Conceptualization:** Shashwat Sridhar, Alexander S. Ecker, Tim Gollisch.

**Data curation:** Shashwat Sridhar.

**Formal analysis:** Shashwat Sridhar.

**Funding acquisition:** Alexander S. Ecker, Tim Gollisch.

**Investigation:** Shashwat Sridhar, Michaela Vystrčilová, Mohammad H. Khani, Dimokratis Karamanlis, Helene M. Schreyer, Varsha Ramakrishna, Steffen Krüppel, Sören J. Zapp.

**Methodology:** Shashwat Sridhar, Mohammad H. Khani, Dimokratis Karamanlis, Matthias Mietsch, Tim Gollisch.

**Resources:** Alexander S. Ecker, Tim Gollisch.

**Software:** Shashwat Sridhar, Michaela Vystrčilová.

**Supervision:** Tim Gollisch.

**Visualization:** Shashwat Sridhar.

**Writing – original draft:** Shashwat Sridhar, Tim Gollisch.

**Writing – review & editing:** Shashwat Sridhar, Michaela Vystrčilová, Mohammad H. Khani, Dimokratis Karamanlis, Helene M. Schreyer, Varsha Ramakrishna, Steffen Krüppel, Sören J. Zapp, Matthias Mietsch, Alexander S. Ecker, Tim Gollisch.

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
