## [Decision Letter · Decision Letter 0]

3 Sep 2025

Modeling spatial contrast sensitivity in responses of primate retinal ganglion cells to natural movies

PLOS Computational Biology

Dear Dr. Gollisch,

Thank you for submitting your manuscript to PLOS Computational Biology. After careful consideration, we feel that it has merit but does not fully meet PLOS Computational Biology's publication criteria as it currently stands. Therefore, we invite you to submit a revised version of the manuscript that addresses the points raised during the review process.

Please submit your revised manuscript within 60 days Nov 03 2025 11:59PM. If you will need more time than this to complete your revisions, please reply to this message or contact the journal office at ploscompbiol@plos.org. Please include the following items when submitting your revised manuscript:

We look forward to receiving your revised manuscript.

Kind regards,

Paul Bays

Academic Editor

PLOS Computational Biology

Marieke van Vugt

Section Editor

PLOS Computational Biology

**Additional Editor Comments:**

Reviewer #1:

Reviewer #2:

**Journal Requirements:**

3) Some material included in your submission may be copyrighted. According to PLOSu2019s copyright policy, authors who use figures or other material (e.g., graphics, clipart, maps) from another author or copyright holder must demonstrate or obtain permission to publish this material under the Creative Commons Attribution 4.0 International (CC BY 4.0) License used by PLOS journals. Please closely review the details of PLOSu2019s copyright requirements here: PLOS Licenses and Copyright. If you need to request permissions from a copyright holder, you may use PLOS's Copyright Content Permission form.

Potential Copyright Issues:

i) Please confirm (a) that you are the photographer of 5B, or (b) provide written permission from the photographer to publish the photo(s) under our CC BY 4.0 license.

4) Please amend your detailed Financial Disclosure statement. This is published with the article. It must therefore be completed in full sentences and contain the exact wording you wish to be published.

1) State the initials, alongside each funding source, of each author to receive each grant. For example: "This work was supported by the National Institutes of Health (####### to AM; ###### to CJ) and the National Science Foundation (###### to AM).".

**Reviewers' comments:**

Reviewer's Responses to Questions

**Comments to the Authors:**

Reviewer #1: This manuscript presents a new approach for characterizing neural feature selectivity by incorporating selectivity to contrast. The method aims to find a compromise between the more complex subunit models and the models based on single stimulus component. The authors show that adding selectivity to contrast improves encoding accuracy under while noise stimulation but not natural stimulation. In the same of natural stimuli, authors note that because they are dominated by low spatial frequencies, there is relative lack of contrast at small scales, leading to the absence of improvement.

To substantiate the main claim that this method offers an improvement relative to subunit models it is necessary to include comparison with at least one of the subunit-based methods for characterizing neural feature selectivity.

Another substantial concern is that perhaps contrast needs to be evaluated not over the receptive field but at a small scales (i.e. subunits) and then combined according to an optimized combination. This would provide a more direct link to subunit models. This also can impact comparison in the case of the natural scenes, because contrast can be evaluated over small scales.

Please note that some subunit models are not technically demanding and can be performed based on spike-triggered covariance.

The paragraph on lines 254-260 about off midget cell type benefiting least from the inclusion of contrast information contradicts the data: technically off parasols benefit the least.

Discussion regarding comparison with natural scenes (Sec. 3.5) is problematic. It implies that subunits are not relevant under natural conditions. But a more likely explanation lies with the suboptimality of the scale over which contrast was estimated, and perhaps also in the temporal domain.

Reviewer #2: The article by Sridhar et al. studied how retinal ganglion cell responses to natural movies of the marmoset ex-vivo retina could be modeled with a spatial contrast (SC) term in a model with few parameters (4). This represents an extension of the work presented by some of the authors in Liu et al. 2022, where they introduced the model to predict ganglion cell responses to static natural images. The study is well written and presented. The figures are clear. As the authors state, this is an incremental work from what has been published, but it was an important step to prove it. They also acknowledge that these are not state-of-the-art models to predict ganglion cell responses, but that they represent a good benchmark, better than the classical LN, which is as easy to fit, yet explains more of the cell activity.

My main remark is that the authors did not explain the reason for switching to marmoset retina, instead of testing their model in salamander and mouse retina as in the previous article. Second, although it is clear that SC is used by the cells more than just mean intensity, the claim that a 4- vs. a 3-parameter model has better performance should be better supported. Did they try any other second-order statistic besides SC (homogeneity, angular second moment, correlation, entropy?) to show it is actually the one encoded by the cells?

Remarks:

1. Figure 2 is technically informative, but it lacks real biological interest. As it is, this figure is not much more informative than the equations. Also us and hs are not defined in the caption.

If the claim is that this cell is responding to high contrast:

◦ Why not superimpose the contrast and the Imean traces calculated on top of the firing rate responses to exemplify that (accounting for latency) the cell responds to high values of one of them?

◦ Why not use a local peak of the contrast or intensity curve to exemplify the stimulus (and convolved stimulus) that the cell is responding to? In fact, this question comes from not knowing if the value of 0.33 is big (or huge) for making the cell respond (or maybe there is some coupling needed between Imean and SC?). Maybe give some reference values? It is very unclear why the values selected are relevant to understand better, or whether they are just random examples.

◦ Then it would be very nice to see two cell examples behaving differently: I would add the OFF midget example here.

◦ Related, line 197: You really need the evolution over time of LSC superimposed on the FR curve to clearly see this relation; and line 203: This is too simplified, we need to see the evolution of the Imean and LSC curves together with FR.

◦ I would also add these curves to Figure 5, maybe not superimposed on the existing curves, but above or below.

2. Line 191: Did you quantify RF size or any other parameter across datasets to discard/find which could be involved in this discrepancy?

3. Line 228: Is the decision to only weight the model with w on LSC based on previous knowledge that there are not pure LSC-coding cells? Or is it possible to have a huge w that disregards Imean? An option to not increase the number of parameters would be to have w1 and w2 with a constraint for them to sum to 1.

4. Line 234: Would the model be even better if you allow some variation of this Gaussian fit? Also, to avoid over-parameterization one may use sigmax and sigmay plus an ellipse angle. I wonder if Imean and LSC may be integrated over different spatial scales. A comment about this is present in line 405, but the justification to argue that they are probably equal is not well explained.

5. Line 324: What do you mean that the models would benefit by properly assessing SC? Was it poorly assessed? Are there better options?

6. One main claim is that they can estimate the spatial scale at which the cells integrate information from the movies. It may be possible to compare this with the scales encountered with static images (taking into account the different species)?

Minor comments:

• It is confusing to cite Fig. 1B before Fig. 1A.

• Line 123: It is not clear if this is known or if it is shown for the first time here. Maybe a citation is needed?

• Line 124: Change “displayed” to “display” (as I understand this is a general remark; if not, rephrase to make it clear it refers to your data).

• Red and black superimposed traces are not colorblind friendly (Fig. 5 and Fig. 6C).

• Line 135: A citation or a reference to the Methods is needed to justify this stimulus.

• Lines 495/496: I think there may be a typo in the second “marmoset”; I think it refers to macaque.

**Have the authors made all data and (if applicable) computational code underlying the findings in their manuscript fully available?**

The PLOS Data policy requires authors to make all data and code underlying the findings described in their manuscript fully available without restriction, with rare exception (please refer to the Data Availability Statement in the manuscript PDF file). The data and code should be provided as part of the manuscript or its supporting information, or deposited to a public repository. For example, in addition to summary statistics, the data points behind means, medians and variance measures should be available. If there are restrictions on publicly sharing data or code —e.g. participant privacy or use of data from a third party—those must be specified.requires authors to make all data and code underlying the findings described in their manuscript fully available without restriction, with rare exception (please refer to the Data Availability Statement in the manuscript PDF file). The data and code should be provided as part of the manuscript or its supporting information, or deposited to a public repository. For example, in addition to summary statistics, the data points behind means, medians and variance measures should be available. If there are restrictions on publicly sharing data or code —e.g. participant privacy or use of data from a third party—those must be specified.requires authors to make all data and code underlying the findings described in their manuscript fully available without restriction, with rare exception (please refer to the Data Availability Statement in the manuscript PDF file). The data and code should be provided as part of the manuscript or its supporting information, or deposited to a public repository. For example, in addition to summary statistics, the data points behind means, medians and variance measures should be available. If there are restrictions on publicly sharing data or code —e.g. participant privacy or use of data from a third party—those must be specified.requires authors to make all data and code underlying the findings described in their manuscript fully available without restriction, with rare exception (please refer to the Data Availability Statement in the manuscript PDF file). The data and code should be provided as part of the manuscript or its supporting information, or deposited to a public repository. For example, in addition to summary statistics, the data points behind means, medians and variance measures should be available. If there are restrictions on publicly sharing data or code —e.g. participant privacy or use of data from a third party—those must be specified.

Reviewer #1: Yes

Reviewer #2: Yes

PLOS authors have the option to publish the peer review history of their article (what does this mean?). If published, this will include your full peer review and any attached files.). If published, this will include your full peer review and any attached files.). If published, this will include your full peer review and any attached files.). If published, this will include your full peer review and any attached files.

...

Reviewer #1: No

Reviewer #2: No

**Figure resubmission:**

**Reproducibility:**



---

## [Decision Letter · Decision Letter 1]

23 Mar 2026

Dear %TITLE% Gollisch,

We are pleased to inform you that your manuscript 'Modeling spatial contrast sensitivity in responses of primate retinal ganglion cells to natural movies' has been provisionally accepted for publication in PLOS Computational Biology.

Best regards,

Paul Bays

Academic Editor

PLOS Computational Biology

Marieke van Vugt

Section Editor

PLOS Computational Biology

Reviewer's Responses to Questions

**Comments to the Authors:**

Reviewer #2: The authors have addressed all of my comments and questions. It is interesting to show that most of the cells of the marmoset retina rely on SC information, which is an important result for the community. The comparison with a subunit model strengthens the computational interpretation. Overall, the study provides valuable insights into neural computations in the retina.

**Have the authors made all data and (if applicable) computational code underlying the findings in their manuscript fully available?**

The PLOS Data policy requires authors to make all data and code underlying the findings described in their manuscript fully available without restriction, with rare exception (please refer to the Data Availability Statement in the manuscript PDF file). The data and code should be provided as part of the manuscript or its supporting information, or deposited to a public repository. For example, in addition to summary statistics, the data points behind means, medians and variance measures should be available. If there are restrictions on publicly sharing data or code —e.g. participant privacy or use of data from a third party—those must be specified.requires authors to make all data and code underlying the findings described in their manuscript fully available without restriction, with rare exception (please refer to the Data Availability Statement in the manuscript PDF file). The data and code should be provided as part of the manuscript or its supporting information, or deposited to a public repository. For example, in addition to summary statistics, the data points behind means, medians and variance measures should be available. If there are restrictions on publicly sharing data or code —e.g. participant privacy or use of data from a third party—those must be specified.requires authors to make all data and code underlying the findings described in their manuscript fully available without restriction, with rare exception (please refer to the Data Availability Statement in the manuscript PDF file). The data and code should be provided as part of the manuscript or its supporting information, or deposited to a public repository. For example, in addition to summary statistics, the data points behind means, medians and variance measures should be available. If there are restrictions on publicly sharing data or code —e.g. participant privacy or use of data from a third party—those must be specified.requires authors to make all data and code underlying the findings described in their manuscript fully available without restriction, with rare exception (please refer to the Data Availability Statement in the manuscript PDF file). The data and code should be provided as part of the manuscript or its supporting information, or deposited to a public repository. For example, in addition to summary statistics, the data points behind means, medians and variance measures should be available. If there are restrictions on publicly sharing data or code —e.g. participant privacy or use of data from a third party—those must be specified.

Reviewer #2: Yes

PLOS authors have the option to publish the peer review history of their article (what does this mean?). If published, this will include your full peer review and any attached files.). If published, this will include your full peer review and any attached files.). If published, this will include your full peer review and any attached files.). If published, this will include your full peer review and any attached files.

...

Reviewer #2: No

---

## [Editor Report · Acceptance letter]

PCOMPBIOL-D-25-00719R1

Modeling spatial contrast sensitivity in responses of primate retinal ganglion cells to natural movies

Dear Dr Gollisch,

I am pleased to inform you that your manuscript has been formally accepted for publication in PLOS Computational Biology. Your manuscript is now with our production department and you will be notified of the publication date in due course.

With kind regards,

Anita Estes
